# Less Precise Can Be More Reliable: A Systematic Evaluation of Quantization's Impact on VLMs Beyond Accuracy

**Aymen Bouguerra** [1]  **Daniel Montoya** [1]  **Alexandra Gomez-Villa** [2,3]  **Chokri Mraidha** [1]  **Fabio Arnez** [1]

## Abstract

Vision-Language Models (VLMs) such as CLIP have revolutionized zero-shot classification and safety-critical tasks, including Out-of-Distribution (OOD) detection. However, their high computational cost hinders efficient real-world deployment. While quantization is a standard solution for efficiency, its broader impact on reliability metrics beyond simple Top-1 accuracy remains critically under-explored. In this study, we conduct a large-scale evaluation of VLM quantization across a comprehensive experimental suite of over 700k evaluation runs with varying configurations. We find that, contrary to the assumption that quantization's noise degrades performance, it can simultaneously improve accuracy, calibration, OOD detection, and robustness to noise, though not to covariate shift or spurious correlations. We leverage these counterintuitive findings to characterize the mechanics of quantization beyond simple regularization: we show that quantization dampens high-rank spectral components, compelling the model to rely more heavily on robust, low-rank features. Ultimately, this spectral filtering effect drives the observed improvements in generalization and noise tolerance, establishing a pathway to deploy faster, more reliable VLMs by utilizing quantization beyond its conventional role. Paper GitHub repository link.

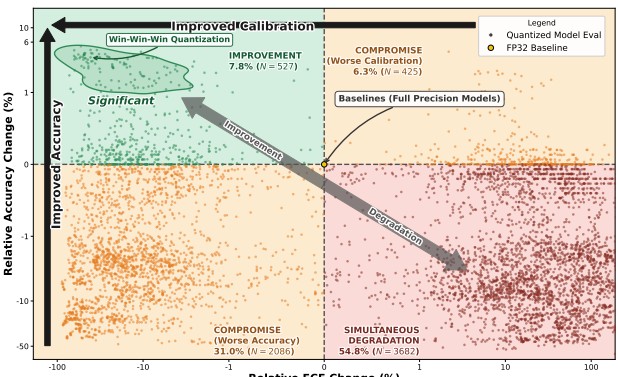

*Figure 1.* Quantization is not systematically destructive; A sweep of 6720 significant W8A8 quantized evaluation runs reveals distinct performance changes relative to FP32 baselines. Surprisingly, nearly 40% of evaluations show improved calibration, including a subset (8.5%) in which both zero-shot accuracy and calibration improve simultaneously. Note that scales are logarithmic.

## 1. Introduction

Vision-Language Models (VLMs), particularly CLIP (Radford et al., 2021), have revolutionized computer vision through their remarkable generalization capabilities. Their powerful zero-shot performance has made them a go-to model for safety-related tasks, particularly out-of-distribution (OOD) detection. And trustworthy AI systems require deep learning models to be developed with a focus on robustness and resilience to ensure reliable and safe deployment (Hendrycks et al., 2021b; Arnez Yagualca, 2023). Consequently, a significant body of work has emerged to rigorously benchmark their reliability, with large-scale evaluations like OpenOOD and ImageNet-X specifically designed to probe the OOD robustness of foundation models (Zhang et al., 2024; Noda et al., 2025; Miyai et al., 2025; Mayilvahanan et al., 2024; Tu et al., 2023).

However, the substantial computational requirements of these models pose a significant barrier to real-world deployment. To mitigate this, model quantization has become the standard compression method, reducing memory and computational overhead by lowering the precision of the model's weights (Courbariaux et al., 2015; Esser et al., 2020). These two domains (reliability and efficiency) have evolved in parallel, creating a critical blind spot. While the community has extensively characterized the intrinsic robustness of full-precision CLIP, it remains largely unknown if these safety-critical properties survive the aggressive, lossy compression of quantization. Our work addresses this gap.

[1]Université Paris-Saclay, CEA, List, F-91120, Palaiseau, France [2]Computer Vision Center, Spain [3]Universitat Autònoma de Barcelona. Correspondence to: Aymen Bouguerra <aymen.bouguerra@cea.fr>.

*Proceedings of the 43rd International Conference on Machine Learning*, Seoul, South Korea. PMLR 306, 2026. Copyright 2026 by the author(s).

Despite quantization's popularity, current evaluation practices suffer from a critical limitation: an overwhelming focus on top-1 accuracy while neglecting essential reliability considerations. The oversight becomes more concerning when compressed models are deployed in real-time systems, where reliability failures would have immediate consequences.

To address this gap, we move beyond conventional accuracy-focused evaluation and ask:

*How does quantization impact VLMs' reliability?*

To answer this question, we systematically evaluate quantization's impact across five critical reliability dimensions: (1) Robustness to quantization noise, examining stability across bit-widths and methods; (2) Uncertainty quality and calibration, evaluating whether models provide reliable confidence estimates; (3) Out-of-distribution detection, evaluating whether the model can distinguish between two or more semantic distributions of samples; (4) Distribution shift robustness, assessing performance under realistic data variations and noise; and finally (5) robustness to spurious correlation, assessing if quantized models rely more or less on spurious correlation for accurate classification.

We evaluate these five dimensions by applying a suite of 16 quantization techniques to 10 pre-trained, non-generative vision-language models[1], and testing them on rigorous benchmarks designed to assess accuracy calibration, OOD detection, and data-shift and spurious robustness. Our analysis reveals interesting findings, partially illustrated in Figure 1, showing that quantization has a measurable impact beyond accuracy, affecting all reliability attributes, indicating a complex interplay between pre-training source and compression strategy. Our findings challenge the view of quantization as a simple compression or regularization tool, and suggest it can be used to improve both model efficiency and reliability. Our major contributions are:

1. **Quantized VLM Reliability Benchmarking:** Through 8k+ quantized models and 700k+ evaluation runs, we uncover a counter-intuitive reliability compromise depending on models: quantization actively improves calibration, OOD detection, and noise robustness by regularizing models with sufficient data redundancy, but degrades data-shifted performance and exacerbates spurious correlations.

2. **Spectral Filtering Mechanism:** We identify the mechanism driving these shifts: quantization acts as a low-pass spectral filter. By injecting destructive noise into the high-rank, low-variance components, it forces models to rely on coarse, robust features, becoming less sensitive to noise and fine-grained semantic nuance.

[1]VLMs (e.g., CLIP) differ from generative, decoder-based LVLMs (e.g., LLaVA)(Miyai et al., 2025)

## 2. Related Work

**Quantization of Pre-trained VLMs.** Quantizing large models involves a trade-off between data-free Post-Training Quantization (PTQ) (Jacob et al., 2018) and the more accurate but data-dependent Quantization-Aware Training (QAT) (Courbariaux et al., 2015). Applying QAT to foundational models requires fine-tuning on training or proxy datasets, creating a significant risk of catastrophic forgetting. While parameter-efficient methods like LoRA (Hu et al., 2022) help, the core tension between adaptation and forgetting remains an open problem.

**Quantization as a Regularizer.** Quantization is increasingly understood as an implicit regularizer (AskariHemmat et al., 2022) that can improve generalization by finding flatter, more robust minima in the loss landscape (Hochreiter & Schmidhuber, 1997; AskariHemmat et al., 2022; Javed et al., 2025). However, this effect has been studied only for weight-only quantization and exclusively through the lens of accuracy and domain generalization. Its impact on a wider range of reliability metrics, and how it interacts with a model's pre-training source (e.g., WIT (Radford et al., 2021) vs. LAION (Schuhmann et al., 2022)), remains largely unexplored.

**Spectral Bias and Feature Granularity.** Deep neural networks exhibit a well-documented "spectral bias," preferentially learning low-frequency, globally coherent functions before fitting high-frequency variations (Rahaman et al., 2019; Xu et al., 2020). In the context of visual representations, this frequency hierarchy maps directly to feature granularity: low-rank spectral components typically encode coarse-grained, robust semantic structures (*e.g.*, global shapes), while the high-rank tail captures fine-grained details, textures, and noise (Yin et al., 2019; Huh et al., 2023; Wang et al., 2020).

**Benchmarking VLM Reliability.** The evaluation of VLM robustness and reliability has matured, with sophisticated benchmarks for OOD detection (OpenOOD) (Yang et al., 2022; Zhang et al., 2024; Wang et al., 2024b) and spurious correlations (CounterAnimal) (Wang et al., 2024b). While these benchmarks have been instrumental in characterizing full-precision models, their application to systematically study the impact of architectural interventions like quantization is nonexistent. Closely related to our work, (Tu et al., 2023) evaluate CLIP's robustness to distribution shifts, OOD detection, and predictive uncertainty. In contrast, in this work, we go several steps further, considering the impact of VLM quantization, a key requirement for computationally efficient deployment.

## 3. Experimental Setup

### 3.1. Architectures and Quantization Protocol

**Models:** We evaluate 10 diverse architectures spanning standard CLIP (ViT and ConvNeXt(Liu et al., 2022)), SigLIP(Zhai et al., 2023), ALIGN(Jia et al., 2021), and CoCa(Yu et al., 2022). We fully describe our experimental setup in Appendix C. To provide a comprehensive view of VLM compression, we evaluate two distinct scopes reflecting different deployment paradigms: (1) **Visual-Only**, where the text encoder remains in FP32. This aligns with standard zero-shot classification, where prompts are static (e.g., fixed class vocabularies). In this regime, text embeddings are typically precomputed and cached offline; thus, the text encoder does not require quantization as it's never reused. (2) **Joint Visual-Text**, where both modalities are quantized to the same precision using the same method. This addresses dynamic open-vocabulary scenarios where prompts are generated at runtime, precluding caching and necessitating efficient real-time inference for both encoders.

**Quantization Strategy:** We utilize simulated quantization across 16 distinct strategies. This includes 8 PTQ methods: Simple MinMax (Jacob et al., 2018), SmoothQuant (Xiao et al., 2023), IGQ-ViT (Moon et al., 2024), QwT (Fu et al., 2025), APQ-ViT (Ding et al., 2022), Rotation-based QuaRot (Ashkboos et al., 2024), Outlier Suppression (Wei et al., 2022), and Q-VLM (Wang et al., 2024a). We further evaluate 8 QAT variants based on Learned Step Size Quantization (LSQ) (Esser et al., 2020) and standard QAT with the Straight-Through Estimator (Courbariaux et al., 2015). For these, we investigate two distillation regimes: *Contrastive-Only* and *Hybrid* (Contrastive + Feature MSE). We also include QAT-LoRA (Xu et al., 2024) and Q-ViT (Li et al., 2022). Post-quantization, we apply Logit Tuning (Radford et al., 2021), optimizing the logit scale on the same proxy calibration set to recover probability calibration. All quantizations are performed on 1000 unique image-caption pairs of CC3M (Sharma et al., 2018), YFCC (Thomee et al., 2016), or SBU (Ordonez et al., 2011).

### 3.2. Evaluation Metrics

To systematically assess reliability beyond standard accuracy, we evaluate across three dimensions. Let $f$ denote the reference FP32 model and $q$ the quantized variant. Let $A(m, \mathcal{D})$ represent the Top-1 accuracy of model $m$ on dataset $\mathcal{D}$.

**1. OOD Detection.** We evaluate the separation between in-distribution ($\mathcal{D}_{\text{ID}}$) and out-of-distribution ($\mathcal{D}_{\text{OOD}}$) confidence scores using **AUROC** and **FPR95**. We utilize 6 scoring functions: 3 traditional (MSP (Hendrycks & Gimpel, 2017), Energy (Liu et al., 2020), Entropy) and 3 VLM-specific methods (MCM (Ming et al., 2022), NegLabel

(Jiang et al., 2024), and EOE (Cao et al., 2024)). Additionally, we report the **relative OOD degradation** ($\delta_{\text{OOD}}$), calculated as the relative change in AUROC between $f$ and $q$ (following the form of Eq. (1)).

**2. Calibration & Robustness.** We measure **Expected Calibration Error (ECE)**(Guo et al., 2017) to assess confidence reliability on $\mathcal{D}_{\text{ID}}$. We also plot enhanced reliability diagrams that show the evolution of samples in the accuracy-confidence landscape to better visualize the impact of quantization. For robustness, we report the relative accuracy change $\delta(\mathcal{D})$ using Eq. (1) under natural shifts (ImageNet-A (Hendrycks et al., 2021c), ImageNet-R (Hendrycks et al., 2021a), ImageNet-V2 (Recht et al., 2019), and ImageNet-Sketch (Wang et al., 2019)) and synthetic corruptions (CIFAR-10-C (Hendrycks & Dietterich, 2019) at Severity 3). Severity values and further details on our full experimental setup are better explained in Appendix C.

$$\delta(\mathcal{D}) = \frac{A(f, \mathcal{D}) - A(q, \mathcal{D})}{A(f, \mathcal{D})} \quad (1)$$

**3. Spurious Impact Metrics.** To isolate the specific bias amplified by quantization from general performance loss, we formulate two derived metrics. These extend standard group disparity measures (Sagawa et al., 2020) and concepts of compression-induced forgetting (Hooker et al., 2019) to quantify relative shifts in spurious reliance. We use these metrics to shift the analytical focus from absolute performance on counter-intuitive samples to the *differential impact of quantization on spurious feature correlation*:

**Delta Relative Spurious Gap ($\Delta$RSG):** Defined as $\text{RSG}(q) - \text{RSG}(f)$, where the Relative Spurious Gap for a model $m$ is $\text{RSG}(m) = \frac{A(m, \mathcal{D}_{\text{N}}) - A(m, \mathcal{D}_{\text{C}})}{A(m, \mathcal{D}_{\text{N}})}$. Here, $\mathcal{D}_{\text{N}}$ and $\mathcal{D}_{\text{C}}$ denote the Natural and Counter-intuitive subsets, respectively. A positive value indicates that quantization disproportionately widens the gap.

**Added Vulnerability ($\text{Vuln}_{\text{add}}$):** Defined as $\delta_{\text{C}} - \delta_{\text{N}}$, where $\delta_{\mathcal{D}}$ is the relative degradation calculated via Eq. (1) for the specific subset. A positive value signifies that the quantized model has degraded more significantly on counter-intuitive examples, implying an increased reliance on spurious features.

## 4. Results

Having established our experimental protocol, we now systematically evaluate quantization's impact across five critical reliability dimensions. We begin by examining zero-shot accuracy to establish quantization viability in Section 4.1, and Figure 16, then proceed to analyze these reliability metrics. In Section 5, we reveal the spectral filtering mechanism that unifies these seemingly contradictory findings.

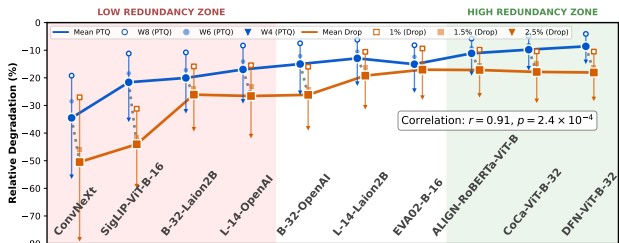

*Figure 2.* Available parameter redundancy indicates PTQ quantizability. A strong correlation between degradation under PTQ (blue) and weight dropout (orange) indicates that redundancy is the primary robustness buffer. Models in the High Redundancy Zone (right) utilize excess representational capacity to absorb quantization noise, whereas Low Redundancy models (left) lack this margin, leading to a significant loss of information.

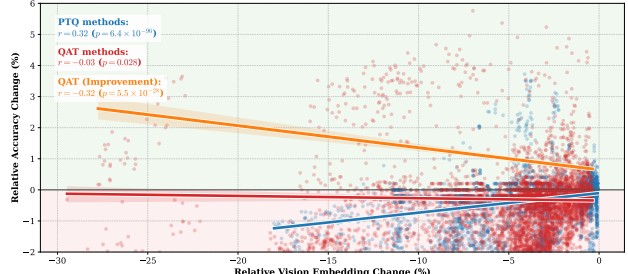

*Figure 3.* Vision embedding similarity between the original and quantized model is a good indicator of information retention for PTQ. However, QAT requires altering embeddings to values that are more quantizable and thus diverge from the baseline embedding.

## 4.1. The Zero-Shot Accuracy Landscape under Quantization

**Less precise can be more accurate.** The results (Figure 1) suggest that weight and activation quantization-aware training (QAT) can aid generalization by driving models toward flatter minima. However, we find that the success of this approach depends heavily on the interplay between data quality and model size. While QAT generally smoothes the optimization landscape compared to the brittle solutions found by Post-Training Quantization (PTQ), a divergence appears in the capacity-constrained `Base-32` architecture (please refer to the Appendix Figure 16 for the per-model quantizability). Here, the model trained on high-quality WIT data quantizes well, whereas the variant trained on noisy LAION data suffers significant degradation. This suggests that the smaller `Base-32` lacks the redundancy to withstand the combination of LAION's inherent label noise and injected quantization noise. In contrast, `Large-14` models don't exhibit this difference, remaining robust and well-quantized across both WIT and LAION datasets, suggesting the model capacity is better for quantizability. Ultimately, reduced precision can improve accuracy depending on how higher and flatter the new minima are. We develop this idea in Appendix H, and plot per-quantization-method results in Appendix Figure 18.

**Data curation and scale generate the redundancy that quantization taps into.** We identify a strong linear correlation ($r = 0.91$) between static quantization degradation and weight dropout sensitivity, confirming that resilience is a function of available parameter redundancy (Figure 2). Crucially, this redundancy is not solely determined by model size, but is heavily modulated by pre-training data quality. Models trained on massive but uncurated, noisy datasets such as `SigLIP` and `CLIP-ViT-B-32-Laion2B` fall into the *Low Redundancy Zone* (left), suggesting that their capacity is fully exhausted fitting the high-entropy

noise of the data, leaving no "safety margin" for quantization. In contrast, models trained on high-quality, filtered data like `DFN-ViT-B-32` inhabit the *High Redundancy Zone* (right), outperforming even larger architectures (e.g., `L-14-Laion2B`). This indicates that data filtering in the DFN model produces cleaner, more redundant parameters, allowing smaller models to survive quantization better than larger models trained on noisy scrapes. The correlation with weight dropout reflects a shared diagnostic (parameter redundancy), not a shared mechanism: dropout is stochastic and full-rank, whereas quantization imposes a deterministic, rank-dependent truncation tied to the discretization grid (Section 5; Appendix Figure 14 contrasts the two spectral signatures directly).

**Fidelity to the baseline is a valid proxy for PTQ success, but a misleading constraint for QAT.** As shown in Figure 3, we measured the visual fidelity by comparing the cosine similarities of the baseline and quantized visual encoders. We observe a statistically significant positive correlation ($r = 0.32$) between vision embedding similarity and accuracy for PTQ methods. This confirms that for post-training quantization, preserving the original manifold is critical; any deviation is effectively "noise" that degrades performance. In sharp contrast, QAT decouples performance from fidelity ($r = -0.03$), demonstrating that fine-tuning allows the model to find new, quantization-friendly minima that are distinct from the FP32 basin. Most strikingly, the subset of models that *improve* over the baseline (Orange Regression, $r = -0.32$) exhibit a significant *negative* correlation; meaning that the highest accuracy gains are achieved not by mimicking the original model, but by actively diverging from its embedding space. This suggests that QAT improves generalizability (Javed et al., 2025) by shedding brittle features. We further address the PTQ vs QAT performance in Appendix F and the counter-intuitive impact of task complexity on quantization in Appendix E.

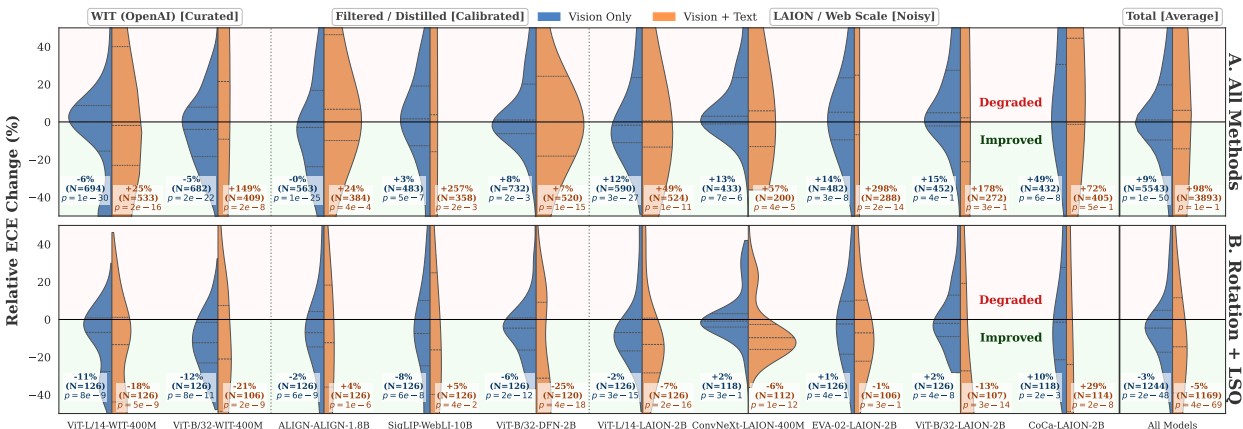

*Figure 4.* Direct impact of successful W8A8 quantizations on calibration. Models are ordered by relative ECE Change. Results indicate a correlation between pre-training data quality and post-quantization calibration degradation: curated sources (WIT) benefit from quantization-induced regularization, while noisy web-data (LAION) leads to increased calibration error. The "Total" distribution demonstrates that vision-only quantization is generally more robust to quantization-induced calibration degradation, whereas text quantization is significantly more brittle. Crucially, comparing Panel A to Panel B confirms that quantization training is necessary to maintain or improve calibration, as PTQ quantization generally renders the model's logit scales obsolete, a problem that is even more pronounced when the text encoder is also quantized.

## 4.2. Calibration and Uncertainty Quality

**Pre-training data quality is correlated to better calibration under quantization.** The impact of quantization on predictive uncertainty (Figure 4) reveals that precision loss does not systematically degrade model trust; its impact is heavily modulated by the pre-training source. Models pre-trained on curated data (*e.g.*, WIT, DFN) frequently exhibit *improved* calibration, as quantization noise provides stochastic regularization that dampens the overconfidence typical of foundation models. Conversely, models trained on noisy web data (LAION) suffer significant degradation (+49% ECE), as precision loss pushes their already diffuse representations into further misalignment. Crucially, we identify text quantization as a primary point of failure, driving a catastrophic +98% increase in calibration error (Panel A). This occurs because zero-shot classification relies on the dot product of two independently quantized manifolds, rendering the pre-trained logit scales obsolete. However, a QAT methods (Panel B) help mitigate this with backpropagation. Effectively absorbing scale distortions and achieving a global -5% reduction in ECE compared to the FP32 baseline, even more so when the text encoder is also quantized.

**Logit Scale Tuning:** This is a form of temperature scaling (Guo et al., 2017) that recalibrates the pre-trained logit scale, rendered obsolete by quantization. To visualize this, we replace static reliability bars with bin trajectories (Figure 5), tracking the evolution of confidence buckets from the baseline. This reveals distinct pathologies: curated models (WIT) start systematically under-confident, while noisy models (LAION) are over-confident. Optimizing logit scales on proxy data while freezing the backbone corrects this ge-

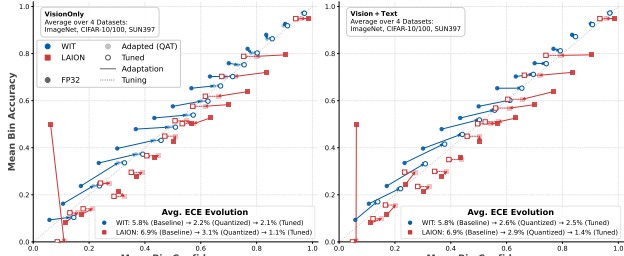

*Figure 5.* **Calibration Trajectories.** We modify standard reliability diagrams to plot the *trajectory* of confidence bins from FP32 (solid) → QAT (faded) → Logit Tuning (hollow). This visualization reveals how quantization and tuning dynamically correct the systematic under-confidence of curated models (WIT, blue) and over-confidence of noisy models (LAION, red), allowing quantized models to surpass FP32 calibration.

ometric mismatch. As shown by the dotted trajectories, this simple scalar realignment effectively pulls predictions toward the diagonal. Consequently, fully quantized models achieve an average ECE of 1.1%, significantly outperforming their own FP32 baselines (6.9%) that have their logit scales trained on the whole pre-trained dataset.

However, the success of this tuning depends sensibly on the quality of the proxy data. As shown in Figure 19 (Left), while accuracy degradation (around 6.5%) remains consistent across proxy sources, calibration is far more sensitive to these changes. The curated CC3M dataset yields a statistically significant improvement in ECE retention ($p < 0.001$) compared to noisy alternatives like YFCC. Furthermore, Figure 19 (Right) reveals a strong "repairability" correlation: models with the most severe post-quantization mis-

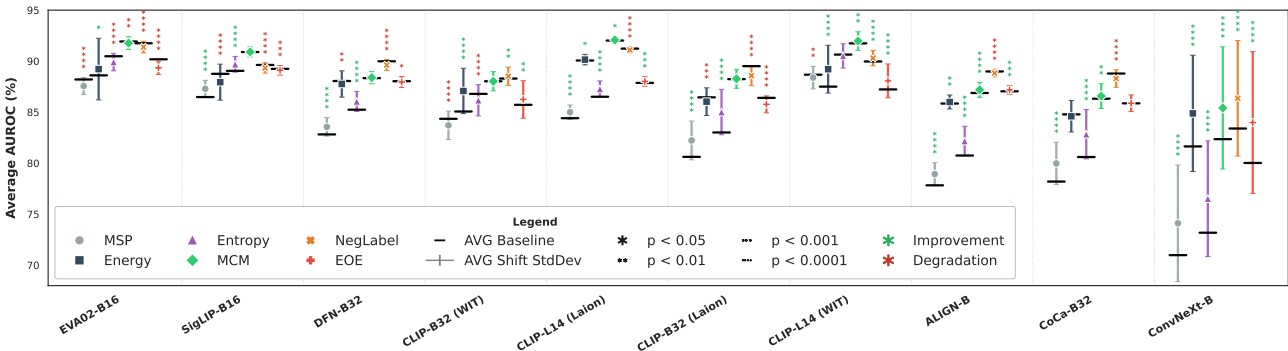

*Figure 6.* Impact of successful W8A8 quantization on OOD Detection (AUROC). Average AUROC across quantization methods (higher is better). QAT methods (center, right) maintain OOD performance for the LAION model, despite this model suffering from significant accuracy and calibration degradation. VLM-specific OOD methods consistently outperform classic methods. Vertical lines represent the maximum improvement and degradation relative to the full precision baseline of that experiment. Refer to Figure 24 for $FPR@95$.

alignment (high ECE after being quantized) experience the largest gains from tuning. This further supports the idea of post-quantization logit obsolescence: as the logit scale is trained for millions of iterations and quantization heavily impacts not only the internal representations but also the model's final accuracy, the logit scale becomes misaligned with the altered distribution of quantized outputs.

### 4.3. Out-of-Distribution Detection: Resilience and Regularization

**OOD In The VLM Era.** We observe a consistent hierarchy where VLM-specific methods leveraging textual anchors (MCM, NegLabel) outperform unimodal baselines (Wang et al., 2023). However, on standard benchmarks, the gap to energy-based scores is narrow, suggesting raw logit norms capture sufficient semantic distance in foundation models. Notably, for architectures such as `SigLIP` and `DFN-B32`, specific quantization configurations yield small but statistically significant AUROC improvements (Figure 6), suggesting that discretization may filter high-frequency noise and clarify decision boundaries. However, since accuracy is also strongly correlated with OOD detection, degradations in the former also impact the latter.

**Re-coupling Accuracy and OOD Detection.** Statistical analysis ($N = 466k$) indicates a strong coupling between classification accuracy and OOD detection. As shown in Appendix Figure 20 (A), metrics are perfectly correlated ($r = 0.994$) (Wang et al., 2024b), this implies that quantization impact on top-1 accuracy and on calibration will also be translated to OOD performance. This coupling reflects a shared sensitivity to representation degradation rather than metric redundancy: PTQ that breaks the logit scale can collapse OOD AUROC while leaving accuracy nearly intact (Figure 4A), and VLM-specific scorers such as NegLabel and MCM condition on top-1 outputs.

**Architectural Divergence:** We observe that `ConvNeXt` benefits significantly more from quantization than Transformer-based architectures. We hypothesize that quantization acts as a spectral filter for CNNs, dampening the high-frequency, textural features that typically lead to OOD overconfidence in full precision. By stripping away this spurious precision, quantization effectively acts as a denoiser, enhancing the separability of the semantic manifold (see Appendix G.1 for detailed separability analysis).

### 4.4. Robustness to Covariate Shift

**The Robustness Dichotomy: Noise vs. Semantics.** As illustrated in Figure 7, quantization introduces a fundamental divergence in how VLMs respond to distribution shifts. We observe a stark contrast between synthetic robustness (resilience to noise, blur, and weather) and semantic robustness (resilience to style changes and natural shifts like ImageNet-A/R). This dichotomy suggests that quantization is not merely a compression operation, but a structural filter that fundamentally alters the model's feature dependence.

**Synthetic Robustness: Quantization as a Low-Pass Filter.** The most significant finding is the universal and massive improvement in synthetic robustness (Right Panel), with an average relative gain of $+8.9\%$ across all architectures. Synthetic corruptions fall into two categories: those that introduce high-frequency noise (e.g., Gaussian noise) and those that remove it (e.g., defocus blur). By discretizing the network weights, quantization effectively limits the model's capacity to represent fine-grained, high-frequency features. This forces the model to rely on coarse, low-frequency spectral components (such as global shape and structure). Consequently, the quantized model becomes robust to high-frequency perturbations, whether they are additive noise (which is ignored) or blur (which removes details that the model relies less on).

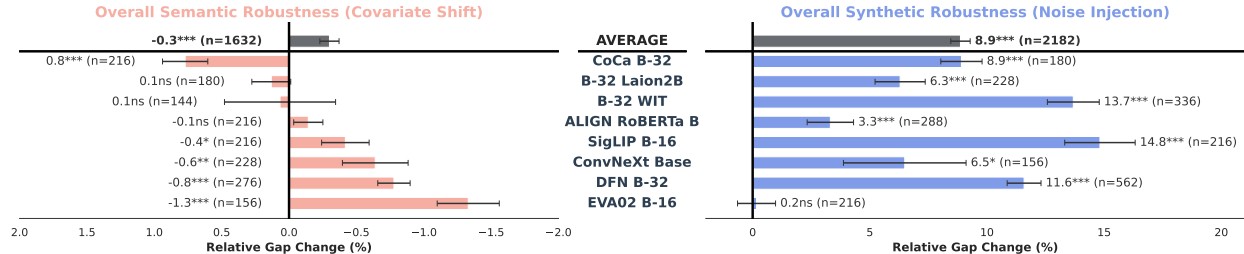

*Figure 7.* Quantization Impacts On Robustness; A comparison of relative robustness gap changes under successful W8A8 quantization (less than 2% degradation on the non-shifted dataset) across varied VLM architectures and quantization methods reveals that while quantization generally slightly degrades semantic robustness (covariate shift, Left), it largely improves robustness against synthetic noise, (Right). Annotations indicate mean change, error, and significance $p < 0.05, 0.01, 0.001$, or non-significance (ns).

*Table 1.* Impact of quantization on spurious correlations (%). We report $\Delta$RSG 3.2 (increase in Relative Spurious Gap) and Vuln. 3.2 (Added Vulnerability). Rot+LSQ does not significantly impact spurious correlations at 8-bit and 6-bit settings. Realized on the Counter-Animal Dataset (Wang et al., 2024b). All results are significant and averaged across 10 independent runs.

| Method | Metric | Bit-width (W/A) | | |
| --- | --- | --- | --- | --- |
| | | **8/8** | **6/8** | **4/8** |
| **Simple PTQ** | $\Delta$RSG | $2.6\pm0.2$ | $2.8\pm0.3$ | $12.5\pm0.6$ |
| | Vuln. | $3.0\pm0.3$ | $3.2\pm0.3$ | $10.3\pm0.4$ |
| **QAT (Contr.)** | $\Delta$RSG | $1.6\pm0.2$ | $2.1\pm0.3$ | $9.1\pm0.5$ |
| | Vuln. | $1.9\pm0.3$ | $2.4\pm0.3$ | $9.1\pm0.4$ |
| **Rot+LSQ** | $\Delta$RSG | $-0.1\pm0.1$ | $0.2\pm0.1$ | $4.0\pm0.3$ |
| | Vuln. | $-0.2\pm0.1$ | $0.2\pm0.1$ | $4.4\pm0.3$ |

**Semantic robustness suffers from feature erasure.** Conversely, this same filtering mechanism proves detrimental under semantic covariate shift (Left Panel). When facing natural distribution shifts involving style transfer or hard examples (ImageNet-A/R/Sketch), we observe a consistent degradation relative to the FP32 baseline. Semantic robustness often relies on fine-grained, high-frequency features to distinguish objects in data-shift contexts; quantization compresses these nuanced signals into shared bins, effectively erasing the discriminatory detail required for these semantic edge cases. Notably, `EVA02 B-16` suffers the most severe degradation, suggesting that architectures optimized for high-density information packing are more susceptible than generative-based architectures like `CoCa`.

### 4.5. Spurious Correlations

**The Bias Amplification Risk.** While quantization improves robustness to synthetic noise, Table 1 reveals a concerning side effect regarding another aspect of reliability. Quantization methods such as PTQ and QAT (families) significantly exacerbate reliance on spurious correlations, increasing the Relative Spurious Gap (RSG) by +2.7% at

W8A8 and a massive +12.5% at W4A8 ($p < 0.001$). This suggests that the filtering mechanism described in the previous section cuts both ways: by dampening fine-grained signals, naive PTQ inadvertently suppresses the subtle invariant features required to distinguish core concepts from their contexts (e.g., distinguishing a "wolf" from a "snowy background"). Consequently, the model reverts to "heuristic" reasoning, latching onto high-magnitude background correlations that survive the discretization process and that are affected by the aggressiveness of the precision constraints.

**Rotation Reduces the Added Spurious Correlations by Quantization.** This bias amplification is not inevitable. The *Rotation + LSQ* method is the most robust quantization technique for mitigating spurious correlation exacerbation, yielding statistically insignificant changes in spurious vulnerability at W8A8 and W6A8 (-0.2 n.s.). By rotating the activation space prior to quantization, this method aligns the semantic manifold with the quantization grid, preventing the "outlier collapse" that typically destroys feature subtleties. However, at extreme compression rates (W4A8), even robust methods succumb to bias (+3.7%), indicating a hard limit beyond which small bit-widths produce large numerical imprecision. This generates increasingly severe quantization noise that erodes the fine-grained features carrying semantic cues separating core concepts from spurious context.

### 4.6. Synthesis of Empirical Findings

We consolidate the headline effects of this section along two independent axes in Appendix Tables 4 and 5. The quantization *method* sets the failure rate and determines whether spurious correlations are amplified (Simple PTQ) or neutralized (Rotation+LSQ); pre-training *data quality* crossed with test-set *granularity* governs the accuracy and calibration outcome; curated, redundant checkpoints (WIT, DFN) evaluated on coarse-grained benchmarks gain the most, whereas noisy web-scale checkpoints (LAION) on fine-grained data degrade.

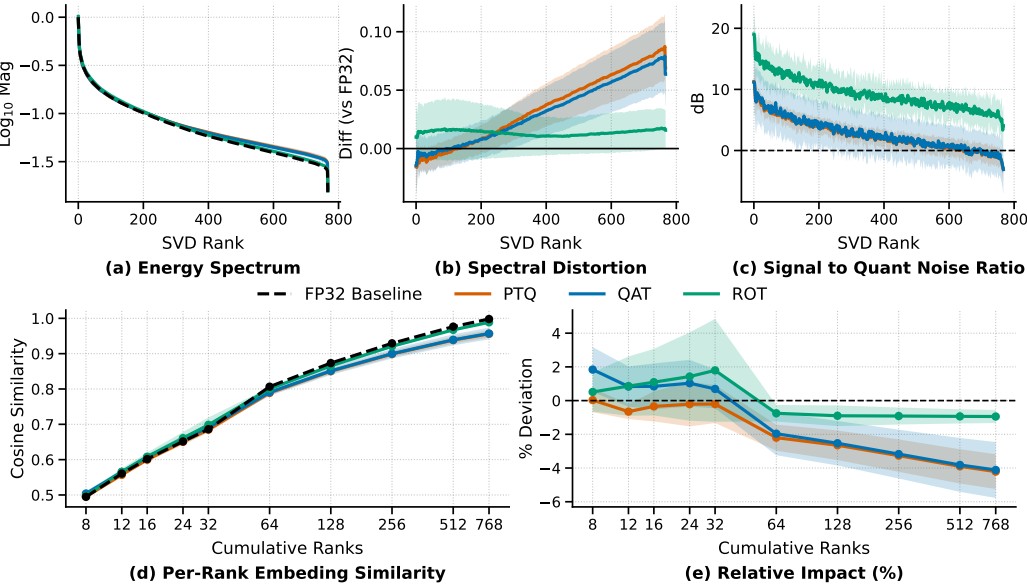

*Figure 8.* Quantization Impact on the Feature Space. (a–c) Spectral analysis reveals that standard quantization disproportionately degrades low-energy components, causing significant spectral distortion and collapsing the SQNR at higher ranks. (d–e) Show how quantization forces the model to do more with less, achieving higher similarity at lower ranks, but worse with more. Conversely, rotation-based methods (ROT) mitigate this increasingly important quantization noise due to their smaller step size, preserving good semantic alignment with the baseline, where standard PTQ and QAT drift significantly at higher ranks.

## 5. Quantization Mechanism: Spectral Filtering

To understand the mechanisms driving the observed trade-offs between reliability metrics, we analyze the spectral properties of the quantized representations. We hypothesize that although quantization injects uniform noise in the value domain, it operates as a non-uniform spectral filter in the feature domain, disproportionately impacting components based on their variance. We performed Singular Value Decomposition (SVD) on the penultimate visual layer of each model in the ImageNet validation set across 3 independent runs, aggregating results across all models and grouping them by method family (PTQ, QAT, or ROT). Our analysis proceeds in two stages: first, projecting quantized features onto the original FP32 basis to quantify signal preservation (Figure 8), and second, recomputing the SVD on the quantized features to observe information reorganization (Figure 9). Please refer to the Appendix D for a detailed methodology description.

**Passive Spectral Filtering.** Figure 8 demonstrates that the Signal-to-Quantization-Noise Ratio (SQNR) decays monotonically with increasing rank. Because the quantization step size is fixed, spectral components with magnitudes comparable to the discretization grid (typically high-rank, low-variance features) are dominated by rounding errors (Figure 8 b-c). This effectively functions as a low-pass filter, preserving the dominant semantic structure while attenuating fine-grained details (Figure 8 d-e). This mechanism

elucidates the synthetic robustness improvements reported in Section 4.4: by truncating the spectral tail where both additive noise and fine-grained degradation manifest, quantization desensitizes the model to these corruptions as it adapts (with QAT) to rely on coarser, generalist features.

**Active Subspace Concentration.** Examining the intrinsic discriminability of the quantized latent space (Figure 9) reveals an active compensation mechanism. In the dominant principal components (Ranks 0–8), quantized models surprisingly achieve higher subspace accuracy than their FP32 baselines (Figure 9A). This suggests a "coarse-grained compensation" effect with QAT: limited representation capacity compels the model to concentrate discriminative information in the most robust, high-variance dimensions, which are less affected by rounding noise. This concentration directly correlates with improved OOD detection, as the suppression of ambiguous, low-variance signals clarifies in-distribution decision boundaries by reducing the noise floor of the representation (see significant gains on ConvNeXt G.1).

**The Semantic Trade-off and Mitigation.** However, this spectral filtering imposes a cost. Figure 9B highlights a significant degradation in accuracy for high-rank subspaces (Rank 64+). Since these dimensions encode the subtle variations necessary for resolving fine-grained classifications and adapting to covariate shifts (e.g., style changes), their suppression explains the reduced performance on semantic

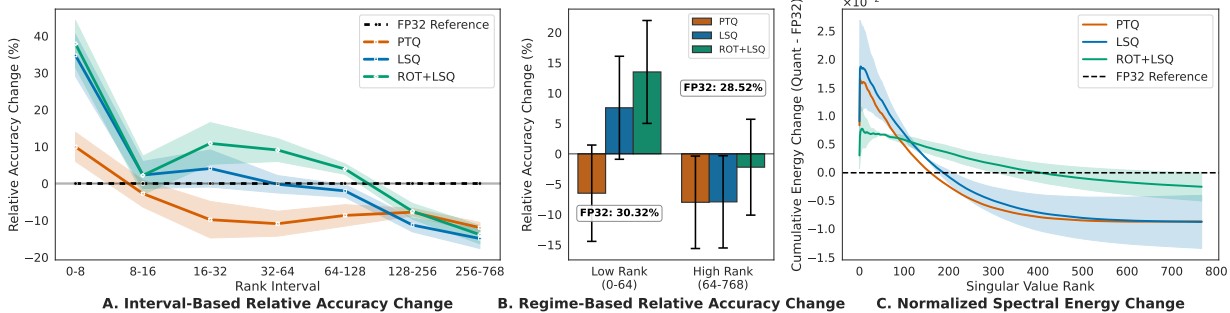

*Figure 9.* Impact of quantization methods on subspace accuracy and spectral energy. (A) Relative accuracy change across rank intervals compared to the FP32 baseline. (B) Direct comparison between low-rank (0–64) versus high-rank (64–768) regimes' accuracy change. (C) Deviation in cumulative spectral energy relative to the FP32 reference, showing energy shift across singular value ranks from higher ranks to lower ranks. Results stem from using 10 full independent runs on `CLIP-ViT-B-32-WIT` on ImageNet-1k.

robustness benchmarks and the increased susceptibility to spurious correlations. Notably, rotation-based methods mitigate this trade-off. As evidenced by the SQNR retention in Figure 8, rotation aligns the activation distribution with the quantization grid. This prevents premature signal decay in the intermediate spectral region (Ranks 8–64), allowing these methods to retain critical semantic detail without sacrificing the regularization benefits of low-rank concentration. Unlike continuous regularizers such as dropout, which inject stochastic full-rank noise and tend to reduce spectral bias, quantization imposes a deterministic, precision-bounded grid on the representation: components whose magnitude approaches the discretization step are dominated by rounding regardless of optimization, which is what amplifies spectral bias rather than reducing it (Appendix Figure 14 compares the two spectral signatures empirically).

## 6. Practical Implications and Deployment

To confirm the simulated reliability gains hold on deployment, we validate our findings by comparing against PyTorch FP32 and a TensorRT FP16 engine on an RTX 4070 ($n$=10 seeds; setup in Section G.5).

*Table 2.* Real TensorRT deployment of CLIP ViT-B/32 (WIT) on CIFAR-100 (RTX 4070, $n$=10 seeds, batch size 64). *Size* is the visual backbone only; text embeddings are pre-computed.

| Precision | Acc (%) | Latency (ms) | Speedup | Size (MB) |
|---|---|---|---|---|
| FP32 (PyTorch) | 64.47 | 120.89 | 1.00× | 302.1 |
| TRT FP16 | 64.43±0.05 | 22.70 | 5.33× | 178.7 |
| TRT INT8 (LSQ) | **66.30±0.39** | **15.66** | **7.72×** | **118.4** |

The INT8 engine *exceeds* FP32 zero-shot accuracy on CIFAR-100 ($+1.83$ pp, $p < 10^{-7}$) while running faster and smaller. Its noise-robustness gains transfer to the deployed engine as well (Table 3): it degrades less than FP16 under

*Table 3.* Blur-robustness of the TensorRT engines (CIFAR-100, $n$=10 seeds; Gaussian kernels $k$=11/23/51 at $\sigma$=2/3.8/8). *Added rob.* is the INT8$-$FP16 gap in accuracy drop (positive: INT8 degrades less); all paired $t$-tests $p < 10^{-8}$.

| Blur | FP16 drop (pp) | INT8 drop (pp) | Added rob. (pp) |
|---|---|---|---|
| Mild | 3.92±0.11 | 1.71±0.36 | **+2.21±0.33** |
| Moderate | 12.28±0.08 | 7.99±0.41 | **+4.29±0.36** |
| Severe | 35.64±0.07 | 30.72±0.64 | **+4.92±0.60** |

Gaussian blur, with the advantage growing with severity. Table 4 generalizes this across method families, identifying Rot+LSQ as the configuration that captures these deployment gains without amplifying spurious correlations.

## 7. Conclusion

This work challenges the prevailing belief that quantization is merely a compromise between efficiency and accuracy. Through the largest systematic evaluation of quantized VLMs to date, we demonstrate that discretization functions as an unintended but powerful spectral regularizer. By suppressing high-rank, low-variance components that are most sensitive to rounding noise, quantization compels the model to rely more on coarse-grained features. When a task requires generalization over specificity, this lower precision can simultaneously improve calibration, uncertainty estimation, and robustness to synthetic noise. However, this same mechanism could erase the fine-grained nuances required for semantic robustness and amplify spurious correlations. Crucially, we identify that the margin between beneficial regularization and destructive collapse is governed by the information quality of pre-training data and the model's capacity, which provides the necessary redundancy to buffer the spectral bias exacerbated by quantization. Overall, this work provides guidance for practitioners and a paramount research direction for utilizing quantization beyond efficiency.

## Acknowledgements

This work was supported by a French government grant managed by the Agence Nationale de la Recherche under the France 2030 program with the reference "ANR-23-DEGR-0001" (DeepGreen).

This publication was also made possible by the use of the CEA-List FactoryIA supercomputer, financially supported by the Ile-de-France Regional Council.

Alexandra Gomez-Villa acknowledges project "PID2022-143257NB-I00", financed by "MCIN/AEI/10.13039/501100011033" and FEDER. This paper was supported by the European Union's Horizon Europe research and innovation program under grant agreement number 101214398 (ELLIOT).

Code will soon be made available in this repository (https://github.com/CEA-LIST/less-precise-more-reliable-vlms).

## Impact Statement

This work provides a systematic evaluation of how model quantization affects the reliability of Vision-Language Models (VLMs) and systems that use them. On the positive side, our discovery that quantization can improve OOD detection and noise robustness offers a pathway to deploying safer AI systems in resource-constrained environments. However, we also reveal a potential ethical risk: the spectral filtering effect can increase vulnerability to spurious correlations and thus amplify unfair and unethical stereotypes that may already be present in the internet-scrapped training sets (Birhane et al., 2021), as it has been previously shown that compression techniques could negatively impact fairness (Hooker et al., 2021). By identifying rotation-based quantization as more robust to this degradation, we provide practitioners with general guidelines for mitigating it. Overall, this paper presents work aimed at advancing the field of Machine Learning, specifically by improving VLM's frugality, which we consider to be a net positive. There could still be more potentially negative societal consequences of our work, none of which we foresee at the moment.

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

# Appendix

## A. Limitations

Our study, while comprehensive, has certain limitations.

- **Simulated Quantization:** Unless explicitly stated otherwise, all experiments utilize simulated (or "fake") quantization. We acknowledge that performance in actual deployment may vary depending on the specific hardware architecture and quantization frameworks employed. We further detail why simulated quantization is imperative in our study in Section G.3, and we verify the validity of all quantization using safeguards detailed in Section G.3. We, however, verified our findings by performing actual quantization and deployment of a CLIP model, as shown in tables 2 and 3, and obtained results similar to those from the simulation.

- **Proxy Dataset:** All quantization methods were adapted to caption-based proxy datasets. We did not explore the effect of quantization of fine-tuned VLMs on downstream tasks. Nor do we include VFMs, LVLMs, or other architectures such as LLaVA and Qwen that build on top of VLMs like CLIP, as that would have uncontrollably expanded the scope of this work.

- **Dynamic Quantization:** In our work, we quantize all linear layers without prior analysis. We did not exclude some linear layers from quantization to gain performance.

- **Pure Zero-shot:** Some important baselines are missing from this work, some due to the complexity of implementation, and others due to relying on ID data, e.g., Mahalanobis (Lee et al., 2018), LoCoOp (Miyai et al., 2023), OmniQuant (Shao et al., 2024).

- **Robustness to Adversarial Attacks:** Although typically categorized under model robustness, we focus exclusively on safety concerns rather than security. Consequently, our scope excludes malicious manipulations and addresses only unintentional, undesirable model behaviors.

- **Generative VLMs:** Our evaluation focuses on discriminative, contrastive-trained VLMs (CLIP and variants). Because CLIP-family encoders serve as the vision backbone of many modern generative VLMs (e.g., LLaVA, Qwen-VL) and recent Vision-Language-Action systems, our findings about how quantization reshapes the encoder's spectral properties and reliability profile should inform decisions about quantizing the vision branch of these larger systems. We did

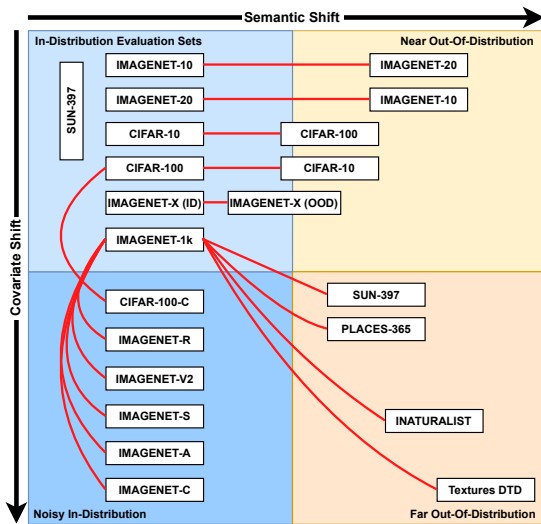

*Figure 10.* Systematic benchmark suite categorized by Covariate and Semantic Shift. Arrows indicate the ID reference dataset for OOD detection.

not, however, directly evaluate downstream generative tasks (hallucination, instruction following, multimodal reasoning), where error accumulation across autoregressive decoding may interact with quantization differently from the discriminative setting. Direct evaluation on generative VLMs is left to future work.

- **Factorial Architecture × Data Design.** While we provide controlled comparisons across model sizes on the same pre-training corpus (ViT-B/32 vs. ViT-L/14 on both LAION-2B and WIT-400M) and across architectures at the Base scale (ViT-B/32, ConvNeXt-Base, EVA02-B-16, SigLIP-B-16), a fully factorial design crossing every architecture with every pre-training data source would require retraining foundation models from scratch, which is beyond our evaluation-focused scope.

## B. Consolidated Effect Summaries

Tables 4 and 5 consolidate the empirical findings of the main paper along two orthogonal axes. Table 4 indexes the W8A8 effects by quantization-method family and includes the observed failure rate; Table 5 crosses pre-training data quality with test-set granularity. Both are qualitative syntheses of the per-experiment results reported in Sections 4.1–4.4: ↑/↓ denote the direction of change relative to the FP32 base-

*Table 4.* W8A8 quantization effects by method family (qualitative synthesis of Sections 4.1–4.4). ↑/↓ denote improvement/degradation relative to the FP32 baseline; = a statistically non-significant change; doubled arrows mark the strongest effects. *Failure rate* is the fraction of evaluations with a > 5% relative accuracy drop (Appendix F). This table is only applicable for easy use cases where we observed gains.

| Dimension | PTQ | QAT | Rotation + LSQ |
|---|---|---|---|
| Accuracy | = / ↓ | = / ↑ | ↑ |
| Calibration (ECE) | ↓↓ | ↑ | ↑↑ |
| OOD detection (AUROC) | = | ↑ | ↑ |
| Synthetic-noise robustness | ↑ | ↑ | ↑ |
| Semantic robustness | ↓ | ↓ | = |
| Spurious correlations | ↓↓ | ↓ | = |
| Behavior at W4A8 (aggressive) | ↓↓ | ↓ | ↓ |
| Failure rate (> 5% acc. drop) | 62.0% | 25.0% | 0.0% |
| Training cost | none | low | low |
| HW kernel support | broad | broad | limited |

*Table 5.* W8A8 quantization effects as a function of pre-training data quality and test-set granularity (qualitative synthesis). Pre-training quality is read from Section 4.1 (parameter redundancy) and the calibration analysis of Section 4; test-set granularity from Appendix Figure 25. Arrows follow the convention of Table 4. The *Synth. robust.* and *Semantic robust.* columns are constant by row, reflecting that those shifts are universal and governed by the spectral filter (Section 5) rather than by the data.

| Pre-training data | Test granularity | Accuracy | Calibration | OOD | Synth. robust. | Semantic robust. |
|---|---|---|---|---|---|---|
| Curated (WIT, DFN) | Coarse-grained | ↑ | ↑↑ | ↑ | ↑ | ↓ |
| Curated (WIT, DFN) | Fine-grained | = | ↑ | = | ↑ | ↓ |
| Noisy web-scale (LAION) | Coarse-grained | = | ↓ | = | ↑ | ↓ |
| Noisy web-scale (LAION) | Fine-grained | ↓ | ↓↓ | ↓ | ↑ | ↓ |

line, = a statistically non-significant change, and doubled arrows the strongest effects in our study. The two tables are complementary: method choice governs the spurious-correlation and failure-rate outcomes, while pre-training data and test-set granularity govern the accuracy and calibration outcomes; the synthetic-/semantic-robustness rows are constant across the data axis, reflecting that those shifts follow from the spectral filter (Section 5) rather than from the data.

### B.1. Deployment Guidelines

The consolidated effects in Tables 4 and 5 translate into a small set of practical recommendations. For deployments where accuracy and inference cost are the primary concern, any QAT family captures the gains reported in Section 6. For safety-critical settings (*e.g.*, medical imaging, autonomous driving), where knowing *when the model is uncertain* matters as much as raw accuracy, Rotation+LSQ is the configuration that simultaneously improves calibration, accuracy, and synthetic-noise robustness without amplifying spurious-correlation reliance at W8A8/W6A8 (Table 1). Naive PTQ is best avoided whenever calibration or spurious-correlation behavior carries a downstream cost; aggressive W4A8 should be reserved for use cases that can tolerate the

failure rates reported in Table 4. Pre-training data quality (Table 5) further modulates these recommendations: curated checkpoints (WIT, DFN) absorb quantization more gracefully than noisy web-scale ones (LAION).

## C. Extended Experimental Setup

This appendix provides the granular configuration details for our large-scale study, which encompasses over 700,000 unique evaluation data points. *Note that we quantized all linear layers, even when they are encapsulated (e.g., in multi-head attention), as well as all convolution layers. The only exceptions to quantization are the absolute last projection layers of each encoder.*

### C.1. Benchmark Dataset Taxonomy

Our benchmark suite systematically covers different forms of distribution shift, and we use the same datasets that are recognized and used by the community (Yang et al., 2024; 2023; Miyai et al., 2025). For more details on dataset preparation for OOD detection, please refer to (Zhang et al., 2024). They are categorized along two dimensions: *Covariate Shift* and *Semantic Shift*, as shown in Figure 10. We also use a specialized dataset for spurious correlations (Wang et al.,

2024b).

## C.2. Proxy Datasets and Training

For methods that require data (QAT, LSQ, and Calibration), we use three primary proxy datasets: **CC3M** (Conceptual Captions), **YFCC**, and **SBU** Captions.

- **Sample Volume:** For each proxy scenario, we stream **1,000 image-caption pairs** from WebDataset shards to serve as the training/calibration distribution. We explain our motivation here G.2

- **Hyperparameters:** Optimization is conducted using AdamW with a base learning rate of $10^{-6}$ for model weights. For LSQ, quantization step-sizes ($\Delta$) are optimized with a higher learning rate of $10^{-4}$ to ensure rapid convergence of the clipping thresholds.

## C.3. Standard and OOD Benchmarks

**Full Evaluation:** All downstream evaluations are performed on the **full validation sets** of each dataset (e.g., all 50,000 images of ImageNet-1k, except for the MOS-subsets to follow their benchmark).

- **Semantic Shift (MOS):** For Far-OOD detection, we utilize the **Maximum Over Softmax (MOS)** benchmark protocol (Huang & Li, 2021). We evaluate ImageNet-1k as the In-Distribution (ID) source against three distinct OOD targets: **MOS-SUN397**, **MOS-Places365**, and **MOS-iNaturalist**, alongside the Describable Textures Dataset (**DTD**).

- **ID-OOD Pairs:** We evaluate 9 specific ID-OOD pairings, including semantic swaps (CIFAR-10 vs. 100) and curated subsets (ImageNet-10 vs. 20 and ImageNet-500 ID vs. OOD).

## C.4. Robustness and Corruption Details

We evaluate robustness using two distinct pipelines:

- **Natural Distribution Shifts:** We utilize the full test suites of ImageNet-V2, ImageNet-A (Adversarial), ImageNet-R (Rendition), and ImageNet-Sketch.

- **Synthetic Corruptions (CIFAR-10-C):** We implement a pipeline using the `imagecorruptions` library, specifically targeting **Severity Level 3** (on a scale of 1–5) to represent significant but non-destructive noise. The corruptions include:
    1. **Gaussian Noise:** Additive electronic noise ($\sigma = 0.08$).
    2. **Defocus Blur:** Simulating camera misfocus via a disk kernel.

3. **Brightness:** Multiplicative intensity shift (factor = 1.5).
4. **Contrast:** Histogram stretching/compression (factor = 1.5).

## C.5. Justifying the 700k+ Evaluation Runs

The combinatorial complexity is calculated as follows:

1. **Model States:** 10 Models $\times$ 3 Precisions ($W8A8, W6A8, W4A8$) $\times$ 2 Scopes (Visual vs. Joint) $\times$ 3 Seeds $\times$ 3 Data Scenarios $\times$ 16 Method Variants $\approx$ **8,640 unique model checkpoints**.

2. **Evaluations per Checkpoint:**
    - **Zero-Shot Accuracy:** 7 datasets x 2 (Logit Tuning) = 14 evals per checkpoint.
    - **OOD Metrics:** 9 Dataset Pairs $\times$ 6 Scoring Functions (MSP, Energy, Entropy, MCM, NegLabel, EOE) = 54 evals per checkpoint.
    - **Robustness:** 4 Natural Shifts + 4 Synthetic Corruptions + 2 Baselines - 2 (ViT-L-14) = 8 evals per checkpoint.
    - **Spurious:** 2 Spurious splits = 2 evals per checkpoint.
    - **Spectral:** 1 dataset (ImageNet) x 10 intervals of ranks x 0.3 (vision-only quantization, 2 seeds instead of 3) = 4 evals per checkpoint.

**Total Calculation:** $8,640$ states $\times$ $82$ evaluations $\approx$ **$708,480$** unique evaluation runs. We will upload all these results in CSV format along with our code. All benchmarks were performed on an Nvidia H200 cluster, and required $\approx 12,000$ GPU-hours.

# D. Spectral Analysis Methodology

To characterize the mechanics of quantization beyond scalar metrics, we developed a two-stage spectral analysis framework. This section details the mathematical formulation for the plots presented in the main text. All analyses were conducted on the penultimate visual features of the ImageNet-1k validation set.

## D.1. Passive Filtering Analysis (Fixed Basis)

This analysis measures how well the quantized model preserves the original information structure. We define $\mathbf{X} \in \mathbb{R}^{N \times D}$ as the activations of the baseline FP32 model and $\mathbf{X}_Q \in \mathbb{R}^{N \times D}$ as the activations of the quantized model.

First, we compute the singular value decomposition (SVD) of the centered baseline features:

$$\mathbf{X} - \boldsymbol{\mu} = \mathbf{U}\boldsymbol{\Sigma}\mathbf{V}^T \tag{2}$$

where $\mathbf{V} \in \mathbb{R}^{D \times D}$ represents the frozen orthogonal basis of the original feature space, and $\mathbf{v}_k$ denotes the $k$-th eigenvector. For Fixed Basis, the SVD ranks are not re-ordered as the energy change when the model is quantized.

**Energy Spectrum & Spectral Distortion.** We project the quantized activations onto the *original* FP32 basis $\mathbf{V}$ to measure the magnitude of the signal retained along each original principal component $k$. The projected magnitude $\sigma_Q^{(k)}$ is given by:

$$\sigma_Q^{(k)} = \sqrt{\frac{1}{N} \sum_{i=1}^{N} \left( (\mathbf{x}_Q^{(i)} - \boldsymbol{\mu}) \cdot \mathbf{v}_k \right)^2} \qquad (3)$$

The **Spectral Distortion** (Figure 8b) is defined as the logarithmic difference relative to the baseline singular values:

$$\Delta\sigma^{(k)} = \log_{10}(\sigma_Q^{(k)}) - \log_{10}(\sigma_{FP32}^{(k)}) \qquad (4)$$

A negative distortion indicates signal attenuation (damping), while positive indicates signal amplification (noise injection).

**Signal-to-Quantization-Noise Ratio (SQNR).** The SQNR (Figure 8c) quantifies the fidelity of the representation for each specific spectral rank. We calculate the Signal Power ($P_S$) and Quantization Noise Power ($P_N$) projected along each eigenvector $\mathbf{v}_k$:

$$P_S^{(k)} = \mathbb{E}\left[ ((\mathbf{X} - \boldsymbol{\mu})\mathbf{v}_k)^2 \right] \qquad (5)$$

$$P_N^{(k)} = \mathbb{E}\left[ ((\mathbf{X} - \mathbf{X}_Q)\mathbf{v}_k)^2 \right] \qquad (6)$$

The SQNR is reported in decibels (dB):

$$\mathrm{SQNR}_k = 10 \cdot \log_{10}\left( \frac{P_S^{(k)}}{P_N^{(k)} + \epsilon} \right) \qquad (7)$$

A collapse in $\mathrm{SQNR}_k$ indicates that the variance of the quantization error exceeds the natural variance of the feature, effectively destroying information at that rank.

**Subspace Similarity Saturation.** To measure global semantic alignment (Figure 8d–e), we reconstruct the embeddings using only the first $k$ principal components and compute their cosine similarity to the full FP32 embedding. Let $\hat{\mathbf{x}}_k$ be the reconstruction at rank $k$:

$$\hat{\mathbf{x}}_k = \boldsymbol{\mu} + \sum_{j=1}^{k} ((\mathbf{X}_Q - \boldsymbol{\mu}) \cdot \mathbf{v}_j) \mathbf{v}_j^T \qquad (8)$$

We report the average cosine similarity: $\mathrm{Sim}_k = \mathrm{CosSim}(\hat{\mathbf{x}}_k, \mathbf{X}_{FP32})$.

### D.2. Active Subspace Analysis (Adaptive Basis)

This analysis investigates how the quantized model *reorganizes* its own latent space. Unlike the passive analysis, we compute a **new** SVD basis $\mathbf{V}_Q$ specific to the quantized model's features:

$$\mathbf{X}_Q - \boldsymbol{\mu}_Q = \mathbf{U}_Q \boldsymbol{\Sigma}_Q \mathbf{V}_Q^T \qquad (9)$$

**Subspace Classification Accuracy.** We isolate specific spectral bands (e.g., Rank $i$ to $j$) to determine which components drive the model's performance (Figure 9A–B). For a given interval $[i, j]$, we reconstruct the features using only the components in that band of the quantized basis:

$$\tilde{\mathbf{x}}_{[i,j]} = \boldsymbol{\mu}_Q + \sum_{k=i}^{j} \left( (\mathbf{X}_Q - \boldsymbol{\mu}_Q) \cdot \mathbf{v}_{Q,k} \right) \mathbf{v}_{Q,k}^T \qquad (10)$$

These partial reconstructions $\tilde{\mathbf{x}}$ are then fed into the model's original classification head (zero-shot text projection) to compute Top-1 accuracy. We report the relative accuracy change:

$$\Delta\mathrm{Acc}_{[i,j]} = \frac{\mathrm{Acc}_Q([i,j]) - \mathrm{Acc}_{FP32}([i,j])}{\mathrm{Acc}_{FP32}([i,j])} \qquad (11)$$

**Normalized Spectral Energy Change.** We analyze the redistribution of information density (Figure 9C). We compute the Cumulative Distribution Function (CDF) of the explained variance for the quantized model:

$$F_Q(k) = \frac{\sum_{i=1}^{k}(\sigma_Q^{(i)})^2}{\sum_{j=1}^{D}(\sigma_Q^{(j)})^2} \qquad (12)$$

The plot displays the differential $F_Q(k) - F_{FP32}(k)$. A positive value indicates that a larger proportion of the model's total energy is concentrated in the earlier ranks than in the FP32 baseline, confirming the model's reliance on the lower ranks. Note that the SVD ranks are recomputed for each model.

### D.3. Causal Validation via SVD Rank Truncation

While the spectral analyses in Sections D and 5 (main text) show that quantization correlates with energy concentration in low-rank components, correlation does not establish a causal chain to the downstream reliability effects we measure. To close this gap, we perform a complementary interventional experiment that explicitly manipulates the spectral content of an FP32 model and re-evaluates each reliability metric on the reconstructed representation. Concretely, for an FP32 visual encoder we compute the SVD of its penultimate-layer activations on the relevant evaluation set, truncate the basis at rank $k \in \{8, 16, 32, 64, 128, 256, 512\}$,

and reconstruct features using only the top-$k$ principal components before passing them through the (unchanged) text-projection head. This replicates the spectral cutoff that quantization implicitly induces, but with $k$ as a directly controllable variable.

**Reliability signature tracks the truncation rank (CIFAR-100, ImageNet-1k).** Figures 11 and 12 report accuracy, calibration (ECE), and Gaussian-blur robustness as a function of $k$. As more of the high-rank tail is removed, the curves reproduce the empirical signature of quantization: accuracy initially holds, calibration improves, and noise robustness increases monotonically before collapsing once $k$ drops below the discriminative core. This monotonic dependence on $k$ is the smoking gun that the spectral filtering mechanism (not a side-effect of finite-precision arithmetic) is what drives the reliability changes.

**Spurious-correlation amplification originates from the loss of high-rank features.** Figure 13 extends the same intervention to the Counter-Animal spurious benchmark. Spurious-feature reliance grows as the high-rank tail is removed, mirroring the $\Delta$RSG / Vuln. amplification reported in Section 4.4 for naive PTQ. This confirms that the spurious amplification is not an idiosyncratic artifact of any specific quantization method — it follows directly from suppressing the high-rank discriminative features that disambiguate core concepts from spurious contexts.

**D.4. Quantization vs. Dropout: A Spectral Comparison**

Section 4.1 reports a strong correlation between a model's degradation under weight dropout and its degradation under PTQ, which we attribute to a shared diagnostic (parameter redundancy) rather than a shared mechanism. Figure 14 makes the mechanistic distinction explicit by comparing the spectral energy shift induced by W8A8 quantization (PTQ and LSQ) against that induced by weight dropout at three rates $(10/20/30\%)$. The $y$-axis is the change in cumulative singular-value energy relative to the FP32 baseline: a positive value at a given rank indicates that more of the representation's total energy has been concentrated into the dominant low-rank components by that point. Quantization concentrates energy in the low-rank head — amplifying the spectral bias by suppressing the high-rank tail — whereas weight dropout, being stochastic and full-rank, produces a markedly different and far weaker spectral shift. The two interventions can therefore correlate at the level of aggregate accuracy degradation while acting through distinct spectral mechanisms, confirming that quantization is not interchangeable with continuous noise-injection regularizers.

**Post-Training Quantization vs. Quantization-Aware Training in VLMs.** Beyond average performance, relia-

bility is defined by worst-case behavior. The Cumulative Distribution Function (CDF) in Figure 15 reveals that PTQ fails the $5\%$ accuracy tolerance threshold in $44.1\%$ of evaluations. Figure 16 offers a per-model landscape. QAT significantly acts as a stabilizer, nearly halving this failure rate to $25.0\%$. As detailed in Figure 18 (see Section F for a full breakdown), this reliability is maximized by the rotation-based method (Ashkboos et al., 2024), which essentially drastically reduces the quantization step by geometrically redistributing the huge magnitude of the outliers (Darcet et al., 2024).

# E. Task complexity imposes redundancy

The statistical landscape in Figure 17 reveals that quantization impact is not uniform but scales with task difficulty. In "easy" regimes (FP32 Accuracy $> 90\%$), decision boundaries are robust enough to absorb precision loss with minimal variance. However, as task complexity increases, the model enters a volatile state where quantization either clarifies or destroys the underlying representation.

**The Regularization Gain:** In low-accuracy regimes, we observe the strongest relative improvements ($r = -0.62$, Left). By reducing precision, we force the model to discard brittle, high-precision details in favor of more stable features that generalize better on difficult benchmarks.

**The Fragility Collapse:** Conversely, these same difficult tasks suffer the steepest degradations when the model's structural margin (Figure 2) is exhausted. For architectures with low native redundancy like `SigLIP`, the noise of quantization becomes a destructive force that wipes out the fragile signal required to solve complex cases, confirming that quantization resilience is a function of both task difficulty and available model capacity.

# F. Granular Analysis of Failure Modes and Catastrophic Forgetting

In this section, we provide a detailed breakdown of the failure rates summarized in Figure 18. We define a "failure" as a relative zero-shot accuracy drop exceeding $5\%$ compared to the FP32 baseline on unseen target datasets. This distinction is critical: because our QAT methods utilize small "proxy" datasets (e.g., CC3M) for calibration, performance on the target benchmarks (e.g., ImageNet) serves as a direct proxy for the model's resistance to **catastrophic forgetting**. **Geometric Alignment Mitigates Outliers without Training Risks.** The dominant trend in our evaluation is the superiority of rotation-based preprocessing. In the PTQ regime (Figure 18, Left), `Rotation PTQ` achieves a failure rate of only $5.6\%$, vastly outperforming `Simple PTQ` ($62.0\%$ failure). By applying random orthogonal matrices to weight and activation spaces, these methods redistribute out-

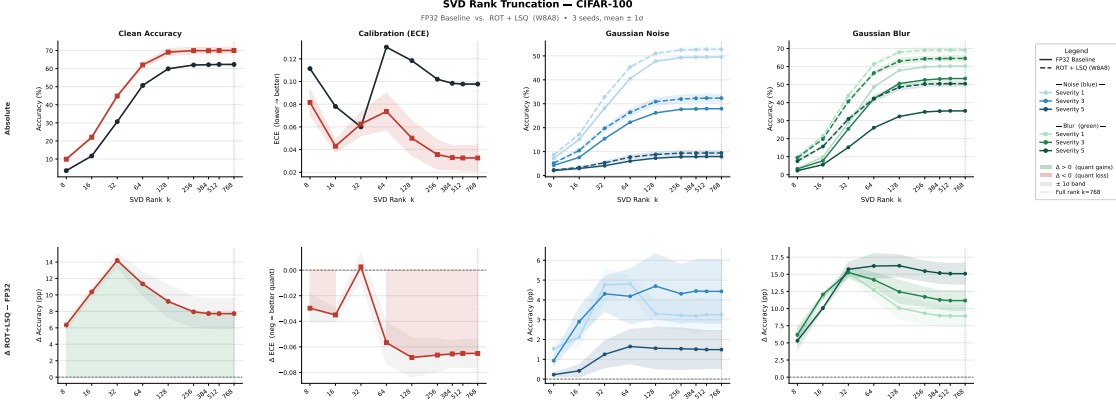

*Figure 11.* SVD rank truncation on CIFAR-100. Reconstructing the FP32 visual representation using only the top-$k$ principal components reproduces quantization's reliability signature: as the high-rank tail is removed, calibration and synthetic-noise robustness improve before accuracy collapses, demonstrating that the spectral cutoff is causally responsible for the reliability effects measured under quantization.

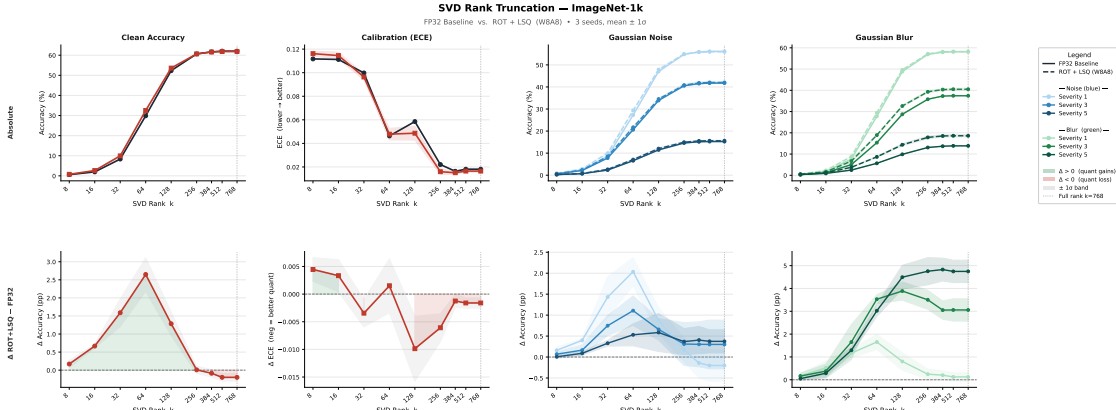

*Figure 12.* SVD rank truncation on ImageNet-1k. The intervention reproduces the accuracy / calibration / noise-robustness trajectory measured under W8A8 quantization, with the effect monotonically growing as more of the high-rank tail is removed.

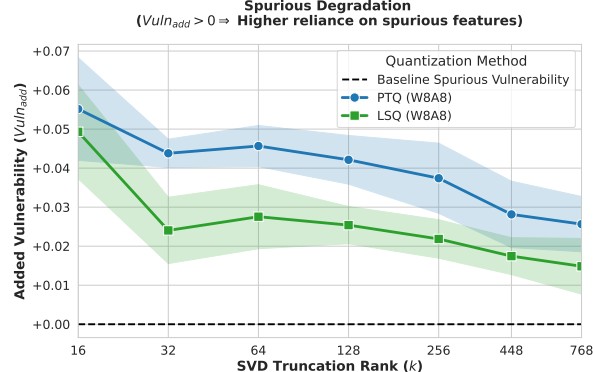

*Figure 13.* SVD rank truncation on Counter-Animal. Spurious-correlation vulnerability tracks the truncation rank, replicating the amplification observed under naive PTQ and confirming that the spurious side-effect originates from high-rank-tail loss, not from quantization-specific noise.

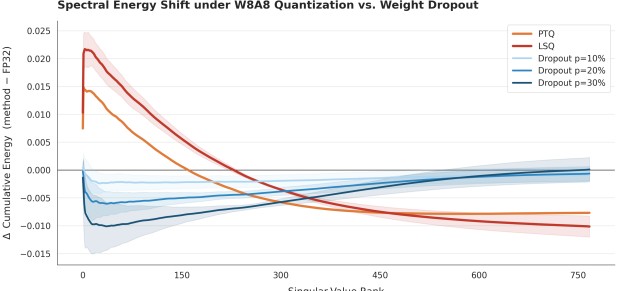

*Figure 14.* Spectral energy shift relative to the FP32 baseline, comparing W8A8 quantization (PTQ, LSQ) against weight dropout at three rates ($10/20/30\%$). The $y$-axis is the change in cumulative singular-value energy ($\Delta$ vs. FP32) along the rank axis. Quantization concentrates energy in the dominant low-rank components (positive shift, amplified spectral bias), while weight dropout exhibits a distinct, weaker signature — confirming that the two operations share a diagnostic (parameter redundancy) but not a mechanism.

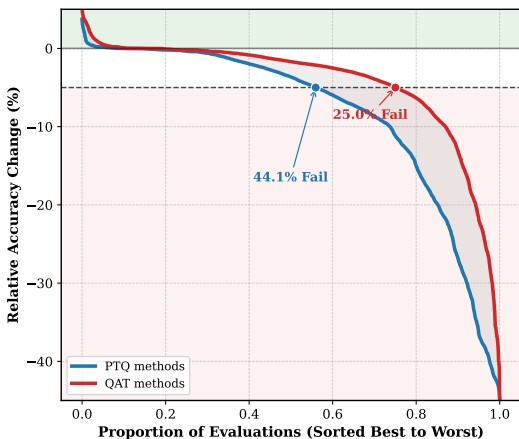

*Figure 15.* PTQ vs QAT overall success (60k runs).

lier features ("massive activations") across multiple dimensions. Crucially, because this operation is strictly geometric and reversible, it renders the model "quantization-friendly" without altering its semantic alignment. This preserves the model's pure zero-shot capabilities, as no gradient updates are applied that could bias the model toward the calibration data. **The LoRA: Parameter Efficiency vs. Spectral Restoration.** A counter-intuitive finding is the high failure rate of `QAT-LoRA` (42.4%), which performs worse than standard full-parameter QAT. While Low-Rank Adaptation is typically prized for preventing catastrophic forgetting, in the context of quantization, it appears structurally insufficient. Quantization noise is high-frequency and distributed across the entire spectral spectrum of the weight matrices. LoRA, by definition, restricts updates to a low-rank subspace. Our results suggest that LoRA lacks the degrees of freedom required to compensate for the pervasive noise of low-bit quantization. Consequently, the adapter overfits to the proxy calibration set in a futile attempt to recover performance, leading to significant degradation on zero-shot tasks (catastrophic forgetting).

**Hybrid Robustness: The Synergy of Rotation and LSQ.**
The most robust configuration observed is `Rotation + LSQ (Hybrid)`, which achieves a perfect 0.0% failure rate. This method combines the best of both worlds:

1. **Rotation** pre-conditions the weight space, flattening outliers and reducing the "quantization error budget" before training begins.

2. **LSQ** (Learned Step Size Quantization) performs minimal, targeted updates to the scaling factors rather than the weights themselves.

By solving the geometric problem analytically (via rotation) and the precision problem parametrically (via LSQ), this hybrid approach minimizes the magnitude of gradient updates required. This drastically reduces the risk of the model drifting away from its pre-trained manifold, thereby preventing catastrophic forgetting and maintaining robust pure zero-shot generalization.

## G. Mechanisms of OOD Separability under Quantization

To explain why OOD detection remains robust despite precision loss, we analyze the distributional dynamics of confidence scores for In-Distribution (ID) and Out-of-Distribution (OOD) samples.

**Quantization-Induced Manifold Contraction.** The kernel density estimates in Figure 20 visualize the impact of quantization on score distributions. In the FP32 baseline, the separability gap between the mean ID and OOD confidence is 1.29. The effect of quantization varies by model stability:

1. **Destructive Regime (Panel B):** When quantization is aggressive or unoptimized, the representation degrades significantly. Both ID and OOD distributions shift toward lower confidence, and their variances overlap, reducing separability.

2. **Stable Regime (Panel D):** In successful W8A8 quantization (defined by $\leq 2\%$ accuracy drop), we observe a *widening* of the separability gap to 1.53.

### G.1. Quantization's Impact on ConvNeXt: A Spectral Perspective

While Vision Transformers (ViTs) exhibit stability under quantization, the `ConvNeXt` architecture demonstrates a distinct positive response, showing a 33.1% relative improvement in OOD separability (Figure 21). Using the spectral framework established in Section 5, we can now attribute this to the interaction between Passive Spectral Filtering and the specific inductive biases of Convolutional Neural Networks (CNNs).

**The Spectral Bias of CNNs.** Unlike the global attention mechanism of ViTs, which naturally promotes low-rank, semantically smoothed representations (shape bias), CNNs exhibit a strong bias toward high-frequency, local textural features. In FP32 precision, `ConvNeXt` utilizes this "spectral tail" (High Ranks >64) to maximize training accuracy. However, for OOD detection, this tail is toxic: it allows the model to assign high confidence to OOD samples based on superficial textural similarities (e.g., a "dog" texture on a non-dog object), resulting in the congested logit space observed in the FP32 baseline (Gap: 1.08).

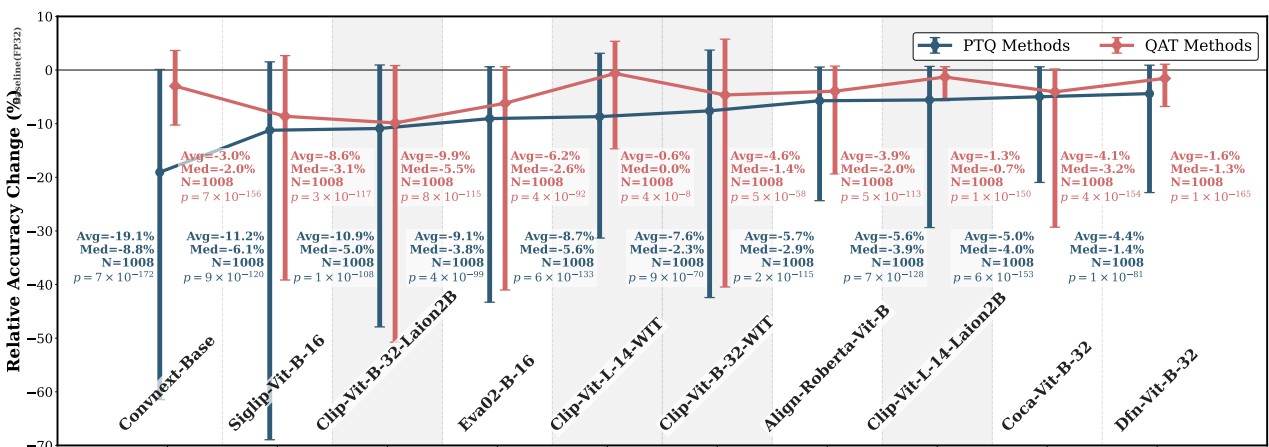

Figure 16. Average impact of post-training and quantization-aware training methods on zero-shot accuracy on several VLMs. Refer to the Appendix Figure 18 for the quantization-method success rate.

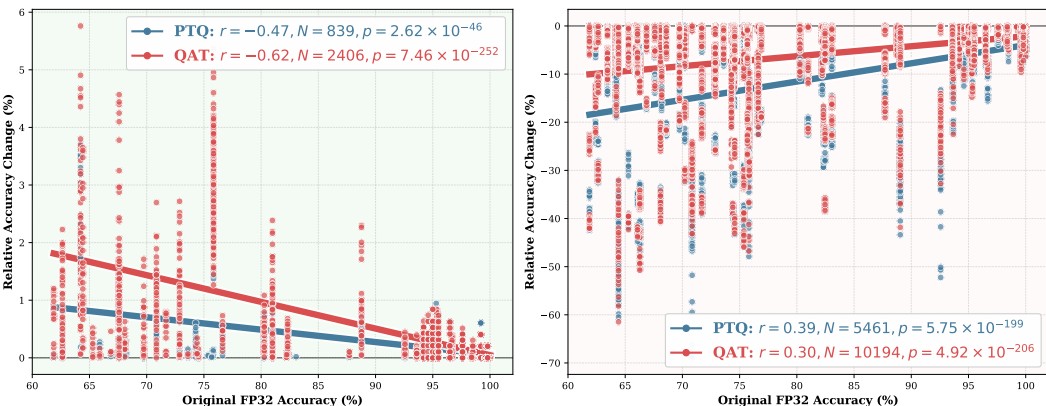

Figure 17. We plot quantization's impact on accuracy relatively to baseline accuracy and separate accuracy improvement (Left) from accuracy degradation (Right), and we notice a strong and significant correlation between the complexity of the task (low zero-shot accuracy) and quantization impact on it.

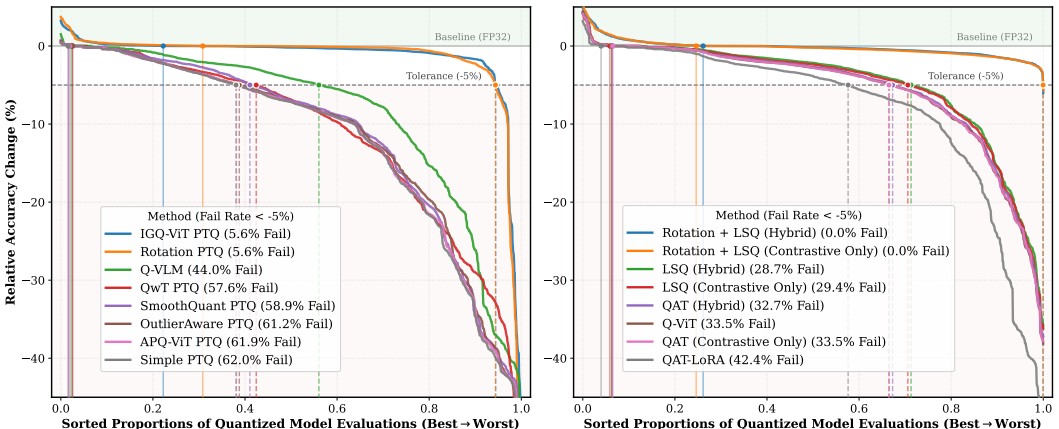

Figure 18. Relative accuracy drops using different PTQ and QAT methods, we consider that a quantization has failed if the relative accuracy drop between the baseline model and quantized one on a test set is higher than 5%.

**Quantization as a Low-Pass Filter.** As demonstrated in Figure 8c, quantization causes a collapse in the Signal- to-Quantization-Noise Ratio (SQNR) for high-rank, low-variance components. For ConvNeXt, this acts as a benefi-

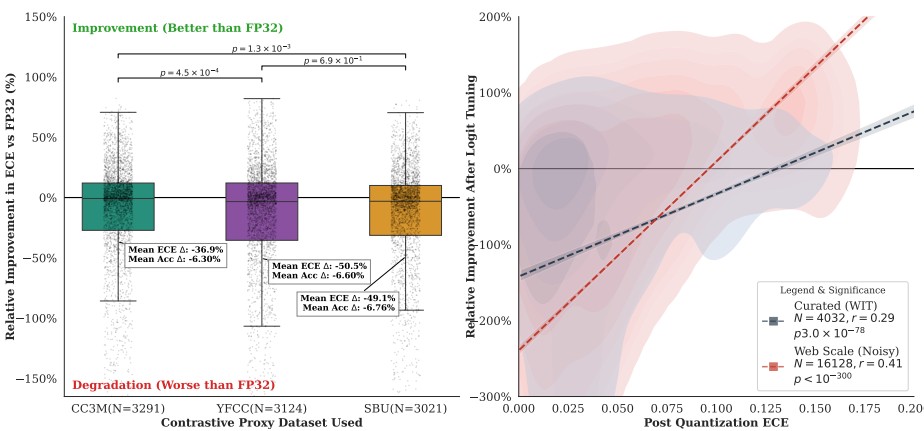

*Figure 19.* **Proxy Data Impact: For Calibration, Data Quality Matters, Unlike Accuracy** (Left) While accuracy is robust to proxy noise, calibration is highly sensitive; the curated CC3M dataset mitigates ECE loss significantly better than noisy alternatives. (Right) Severe post-quantization misalignment yields proportionally greater gains from logit-scale tuning, confirming that reliability can be repaired via logit-scale tuning.

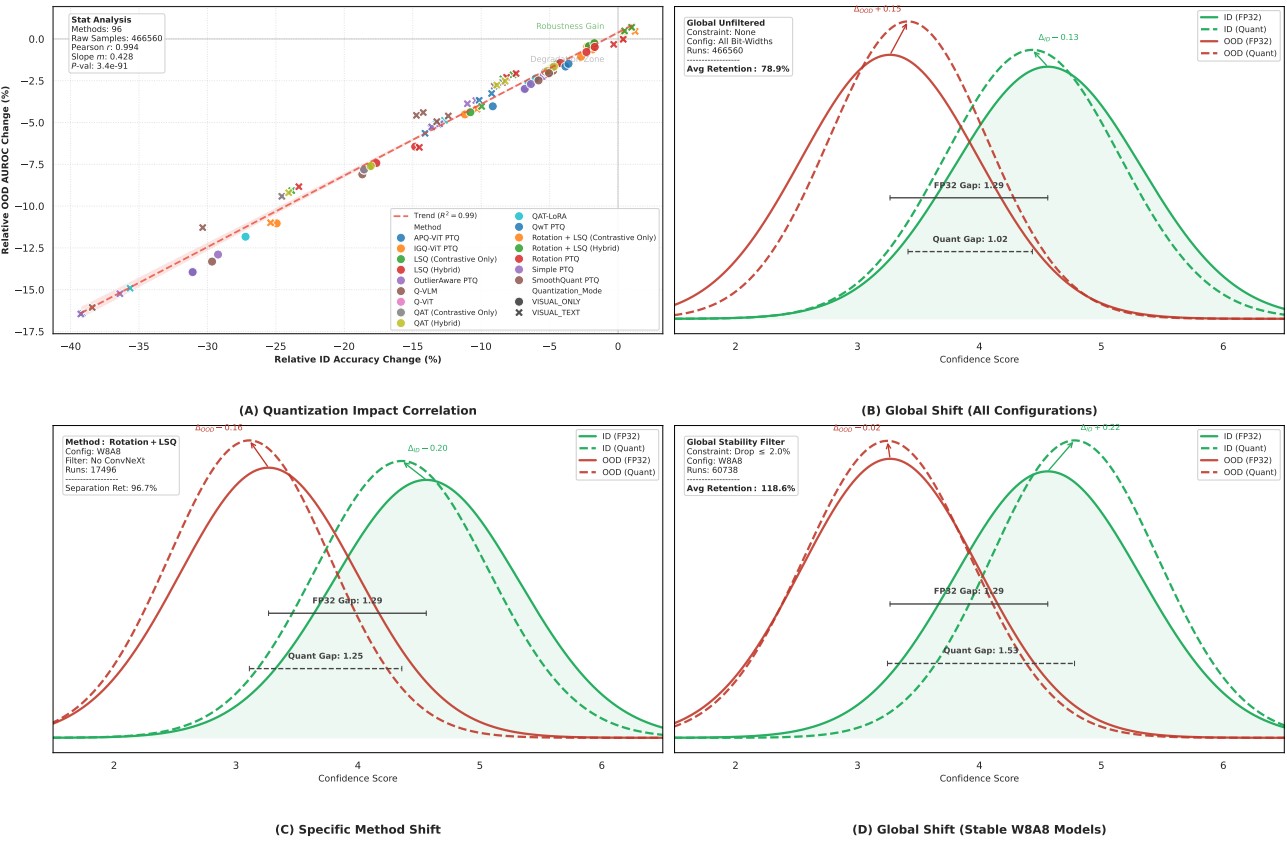

*Figure 20.* **Mechanics of Quantization-Induced Separability.** (A) Global correlation analysis ($N = 466k$) reveals a shallow slope ($m = 0.428$), indicating OOD detection is more robust than accuracy. (B-D) Kernel Density Estimates of confidence scores. While aggressive quantization causes distribution collapse (B), for (C), the preservative power of rotation push both the ID and OOD distribution as quantization noise is added, maintaining separability as accuracy/confidence degrades, while stable W8A8 models (D) exhibit "Manifold Contraction," where ID samples anchor to quantized centroids while OOD samples remain low-confidence, effectively increasing the separability gap ($\Delta_{Gap}$) from 1.29 to 1.53.

cial **spectral filter**:

1. **Tail Truncation:** The quantization grid effectively acts as a low-pass filter, mathematically preventing the

*Figure 21.* **Quantization as a Spectral Filter for ConvNeXt.** Visualization of the aggregated In-Distribution (ID) and Out-of-Distribution (OOD) score distributions for `ConvNeXt-Base`. Unlike standard compression degradation, quantization increases the ID-OOD separability gap by **33.1%** relative to the FP32 baseline (Ghost lines). The *Spectral Filtering* mechanism (Section G.1) effectively dampens the high-frequency "spurious precision" (Red arrow) that drives OOD overconfidence in FP32, while the robust low-rank core anchors the ID confidence (Green arrow), verifying the hypothesis that discretization acts as a beneficial denoiser for CNNs.

model from encoding the fine-grained, high-frequency textural features that drive OOD overconfidence.

2. **Forced Shape Bias:** With the "textural shortcut" removed, the model is compelled to rely on the **Active Subspace Concentration** described in Figure 9A. The decision boundary shifts to rely almost exclusively on the robust, low-rank semantic core (Ranks 0-8), which encodes global shape and semantic structure.

**Resulting Topography.** As visualized in Figure 21, this spectral purification increases the separability gap to 1.43. By stripping away the high-rank noise that CNNs are prone to overfitting, quantization ironically acts as a regularizer that aligns the ConvNeXt latent space more closely with the robust, low-rank topology typical of ViTs.

### G.2. Quantization Sample Size Selection

To determine the optimal calibration dataset size for our quantization pipeline, we performed a scaling study ranging from 1 to 10,000 unique samples across three proxy datasets (CC3M, YFCC, SBU). As illustrated in Figure 22, our analysis identifies $N = 1000$ as the critical sample budget.

In the low-data regime ($N < 1000$), standard methods like LSQ suffer significant degradation (up to -4% accuracy drop on ImageNet), as they fail to estimate stable step-size parameters from sparse statistics. Conversely, extending the calibration set beyond this point ($N > 1000$) yields dimin-

ishing returns and introduces the risk of catastrophic forgetting. Because we employ Quantization-Aware Training (QAT) on proxy data that is distributionally distinct from the target benchmarks, excessive training on large proxy subsets causes the model parameters to overfit the proxy domain, drifting away from the generalizable features of the pre-trained FP32 manifold. Therefore, $N = 1000$ represents the requisite trade-off: it provides sufficient diversity for the convergence of data-hungry methods like LSQ while preventing the distributional drift associated with large-scale proxy fine-tuning.

### G.3. Simulated Quantization

A crucial aspect of our methodology is the use of simulated (or 'fake') quantization. This approach is strictly necessary for several converging reasons. First, hardware limitations: Current GPUs lack native arithmetic support for the non-standard bit-widths we evaluate (e.g., INT6, INT4) and often lack support for fully quantized activations in Transformer architectures, for our comparisons to be valid, they require to apply the same quantizations to all models and architectures, which would be impossible without simulations, as not all architecture-specific layers have quantized kernels. Second, isolation: Simulation allows us to isolate the theoretical impact of precision loss (spectral filtering) from hardware-specific implementation quirks (e.g., kernel overflow or accumulation limits, layers fusion, etc..), ensuring our findings regarding reliability are fundamental to the compression method rather than the hardware backend.

*Figure 22.* Sensitivity Analysis of Calibration Sample Size: We evaluate the impact of the number of unique proxy samples ($N \in [1, 10^4]$) on Zero-Shot Accuracy relative to the FP32 baseline. The shaded regions represent the 95% confidence interval over 5 random seeds. While *Rotation+LSQ* (Orange) demonstrates high robustness at low sample regimes, standard *LSQ* (Blue), while surprisingly effective with the statistics of a single image, continues to converge when fed with more unique samples. We select $N = 1000$ as the optimal operating point to balance quantization effectiveness against the risks of overfitting to the proxy distribution and catastrophic forgetting (Kumar et al., 2022).

Finally, while libraries like bitsandbytes offer efficient kernels for weight-only quantization, they do not support the comprehensive weight-and-activation quantization required to model the full spectral regularization effects observed in this study.

**Verification of Simulation Correctness.** To ensure our simulation was valid, we implemented a verification utility. For any quantized model, this utility counts the number of unique values in both the weight and activation tensors (via forward hooks). For a successful INT8 simulation, this count must be $\leq 2^8 = 256$, which we confirmed for our implementations.

### G.4. Motivation: Why Not Use Existing Kernels like QLoRA, bitsandbytes for Inference?

While methods like QLoRA enable the *training* of models with 4-bit weights, they are primarily designed to reduce memory usage during the training phase, not to accelerate inference. As shown in our direct performance benchmark in Table 6, using these kernels for inference can lead to a significant *slowdown* compared to the FP32 baseline. The overhead of de-quantizing the weights on-the-fly for each computation outweighs the benefits of reduced memory bandwidth. True inference acceleration requires dedicated hardware and software support for low-precision matrix multiplications for multi-head attention, which is what our simulation-based study aims to evaluate in terms of potential accuracy and reliability before such support becomes widespread.

*Table 6.* Performance and memory benchmark of OpenAI CLIP models using non-simulated, kernel-based quantization (bitsandbytes). Inference time is the average latency per batch on an RTX 3090 GPU. Note the significant slowdown for 8-bit and 4-bit inference, which motivates our use of simulation to evaluate accuracy/reliability trade-offs, as current kernels are not optimized for inference speed. This table highlights why we must use simulated quantization and not confuse general quantization frameworks like QLoRA, bitsandbytes that quantize to train, with our aim of lightweight deployment.

| Model | Prec. | Mem. | Time | Speed | Gain |
|---|---|---|---|---|---|
| | | (MB) | (ms) | ($\times$) | (%) |
| B/32 | FP32 | 358.3 | 19.6 | 1.00 | 0.0 |
| | FP16 | 297.8 | 17.7 | 1.11 | 16.9 |
| | INT8 | 179.8 | 81.1 | 0.24 | 49.8 |
| | NF4 | 123.8 | 47.7 | 0.41 | 65.4 |
| B/16 | FP32 | 579.5 | 12.3 | 1.00 | 0.0 |
| | FP16 | 294.6 | 14.2 | 0.87 | 49.2 |
| | INT8 | 177.7 | 90.9 | 0.14 | 69.3 |
| | NF4 | 119.7 | 49.8 | 0.25 | 79.3 |

### G.5. Real-Engine Validation with TensorRT INT8 — Setup and Coverage

The headline accuracy, latency, size, and blur-robustness numbers for this deployment are reported in the main paper as Tables 2 and 3 (Section 6); this appendix documents the conversion setup and coverage notes that did not fit in the main text.

**Setup.** We converted CLIP ViT-B/32 (WIT) into a TensorRT INT8 engine with explicit Q/DQ node placement

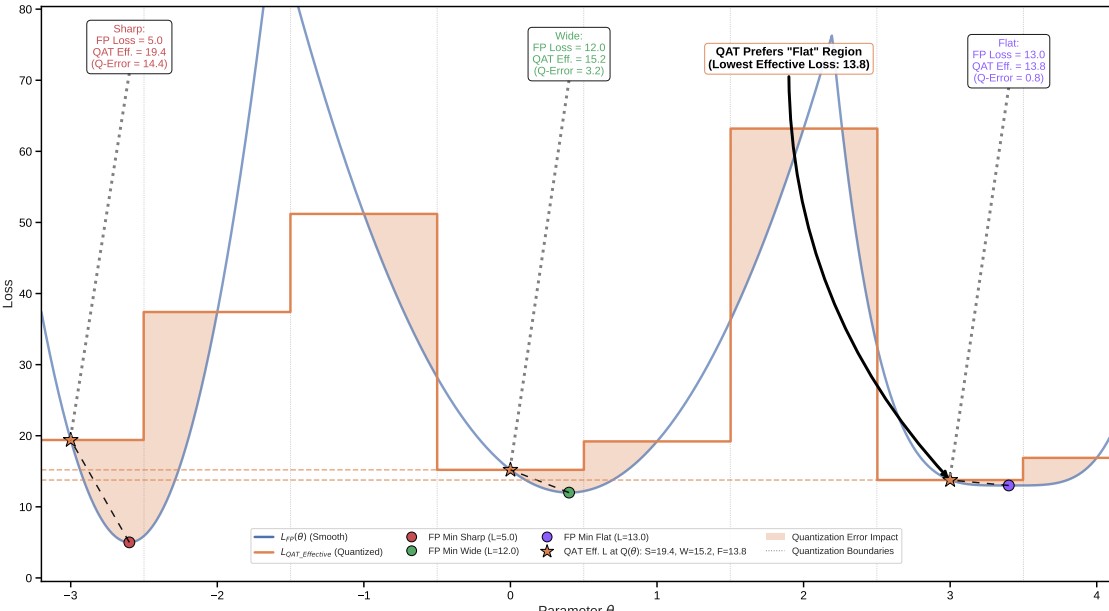

*Figure 23.* A conceptual illustration of how QAT forces the optimizer to abandon a sharp minimum in favor of a flatter, more robust solution that has less loss under quantization.

(LSQ-trained scales) and benchmarked it against PyTorch FP32 and a TensorRT FP16 engine on an NVIDIA RTX 4070 (CUDA 12.4, TensorRT 10.5). Latency is measured with synchronized CUDA events over 1,000 warm batches of size 64 after 100 warm-up batches. FP16 inference matches FP32 accuracy within numerical noise on the clean set, so the relevant comparison for the INT8 engine is against FP16. All numbers in Tables 2–3 are averaged over $n = 10$ independent seeds.

**Coverage and reproducibility.** Converting a fake-quantized model into a TensorRT engine while preserving its Q/DQ scheme requires substantial architecture-specific engineering, and CLIP W8A8 support in TensorRT was added only recently. We deployed a representative configuration (LSQ on ViT-B/32) to validate that the simulated findings transfer; broader engine-level deployment of every quantization method and architecture is beyond the scope of this study and left to future work. The full conversion and benchmarking pipeline (including `lsq_cifar100_trt.py`) is released alongside our code.

### G.6. The core functioning of Fake Quantization

Fake quantization is a simulation technique that models the error introduced by quantization and de-quantization within a standard full-precision (FP32/FP16) training and inference loop. The process for a tensor $x$ is as follows:

1. **Quantize:** The full-precision input tensor $x$ is scaled, shifted, and rounded to the nearest integer value within

the target bit-width's range (e.g., [-128, 127] for INT8).

$$x_{quant} = \text{round} \left( \frac{x}{\text{scale}} + \text{zero\_point} \right)$$

2. **Clamp:** The integer values are clamped to the representable range of the target bit-width.

3. **De-quantize:** The clamped integer tensor is immediately converted back to a full-precision floating-point tensor.

$$x_{dequant} = (x_{quant} - \text{zero\_point}) \times \text{scale}$$

The resulting tensor, $x_{dequant}$, has the same data type as the input but contains the precision loss that *would have occurred* in a true low-bit system. For the backward pass, a **Straight-Through Estimator (STE)** (Courbariaux et al., 2015) bypasses the non-differentiable 'round' function, enabling the model to learn weights that are robust to the simulated quantization noise.

## H. On QAT's Preference for Flat Minima

A significant body of work establishes that neural networks converging to "flat" minima in the loss landscape generalize better (Hochreiter & Schmidhuber, 1997). More recently, (Javed et al., 2025) has shown that weight-only quantization-aware training can improve domain generalization by leveraging this phenomenon. As illustrated in Figure 23, QAT inherently penalizes sharp regions because the weight perturbation from quantization ($Q(w) - w$) causes a large increase

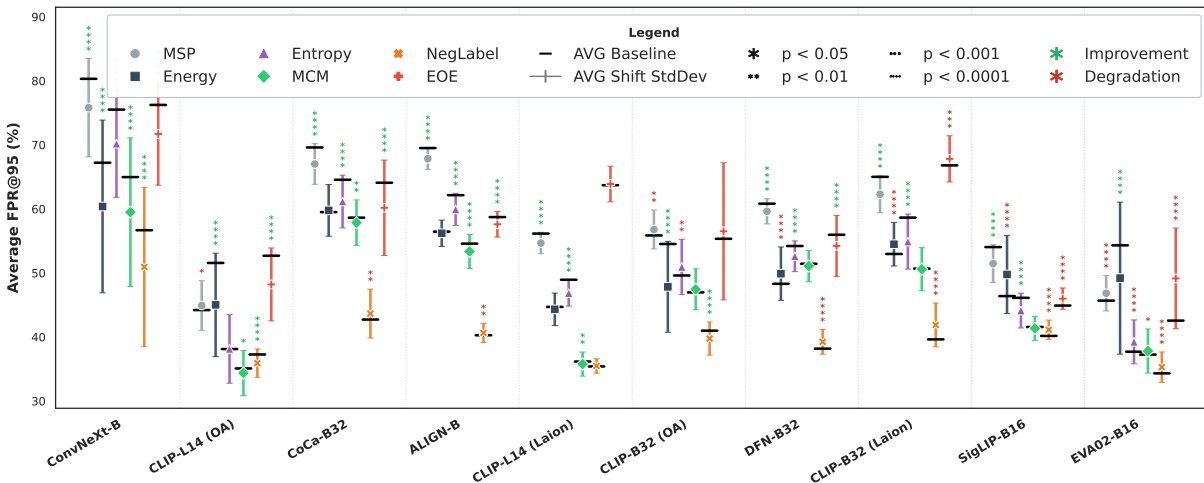

*Figure 24.* Impact of successful W8A8 quantization on OOD Detection (FPR@95). Average FPR@95 across quantization methods (lower is better). QAT methods (center, right) maintain OOD performance for the LAION model, despite this model suffering from significant accuracy and calibration degradation. VLM-specific OOD methods consistently outperform classic methods. Vertical lines represent the maximum improvement and degradation relative to the full precision baseline of that experiment.

in loss. This forces the optimizer to seek out flatter, more robust regions, which can sometimes lead to better generalization and accuracy than the original FP32 model. To be comprehensive, we illustrate the discrepancy between the theoretical full-precision optimization landscape, denoted by the continuous curve $\mathcal{L}_{\mathrm{FP}}(\theta)$, and the effective loss realized under discrete quantization constraints, represented by the stepped function $\mathcal{L}_{\mathrm{QAT}}(\theta)$. The landscape features three distinct minima characterized by varying local curvature (Hessian magnitude): a "Sharp" global minimum, a "Wide" intermediate basin, and a "Flat" local minimum. While the Sharp region ($\theta \approx -2.8$) achieves the lowest theoretical loss ($\mathcal{L}_{\mathrm{FP}} = 5.0$), its high curvature renders it hypersensitive to the parameter perturbations $\delta$ introduced by the quantization grid, resulting in a substantial discretization error ($Q$-Error $\approx 14.4$). In contrast, the "Flat" region ($\theta \approx 3.5$), despite a higher baseline loss ($\mathcal{L}_{\mathrm{FP}} = 13.0$), exhibits superior robustness to quantization noise. Due to the low gradient magnitude $\nabla_\theta \mathcal{L}$ and minimal curvature in this region, the quantization grid aligns more favorably with the loss surface, yielding the lowest *effective* quantized loss ($\mathcal{L}_{\mathrm{QAT}} = 13.8$). This demonstrates that quantization-aware training implicitly favors flat minima, as they provide the necessary tolerance for the discrete mapping of parameters where $\mathcal{L}(\lfloor\theta\rceil) \approx \mathcal{L}(\theta)$.

**CLIP Quantization and Fairness** Previous work on CLIP's quantization (Slyman et al., 2024) has revealed small but generally non-significant increases in unfair biases. Note that they infer using non-deployable quantization frameworks, such as "HuggingFace 4-bit Quantization," which uses QLoRA (quantization for training larger models with less memory) but does not increase inference speed (as

tested in Table 6) and does not quantize activations; "Pytorch 8-bit Quantization," which only supports CPU PTQ methods; and "LLM.int8()", which does not quantize activations. We highlight the need for engineering low-precision kernels for CLIP that truly aim for lightweight deployment.

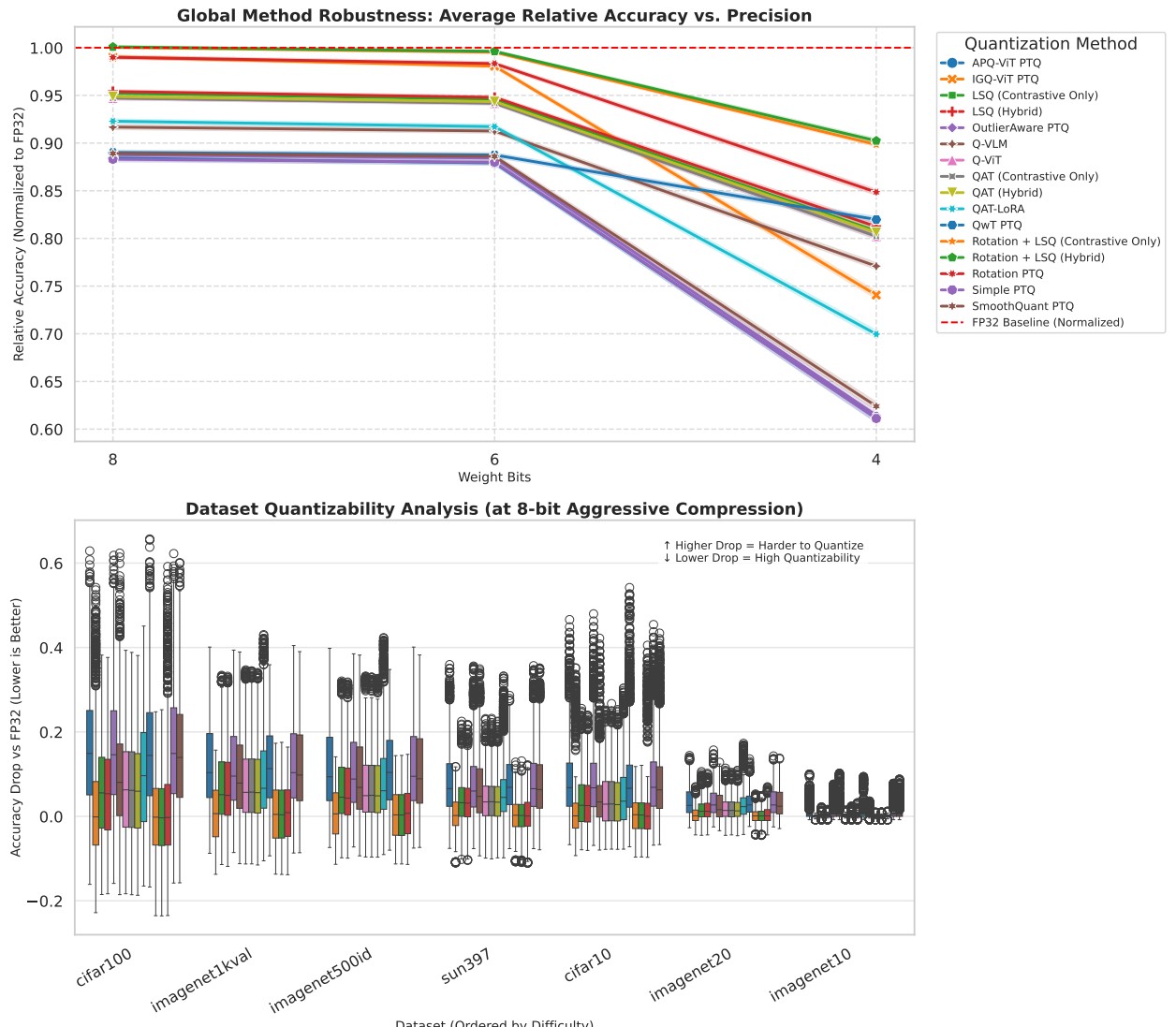

*Figure 25.* (Top) Global accuracy retention per quantization method relative to increasingly aggressive quantization. (Bottom) Per the downstream dataset, zero-shot evaluation after quantization (aggregation of 60k data points) to show that low-resolution datasets like CIFAR-100 benefit more because they are more coarse-grained and feature-friendly. (0.6=60%)

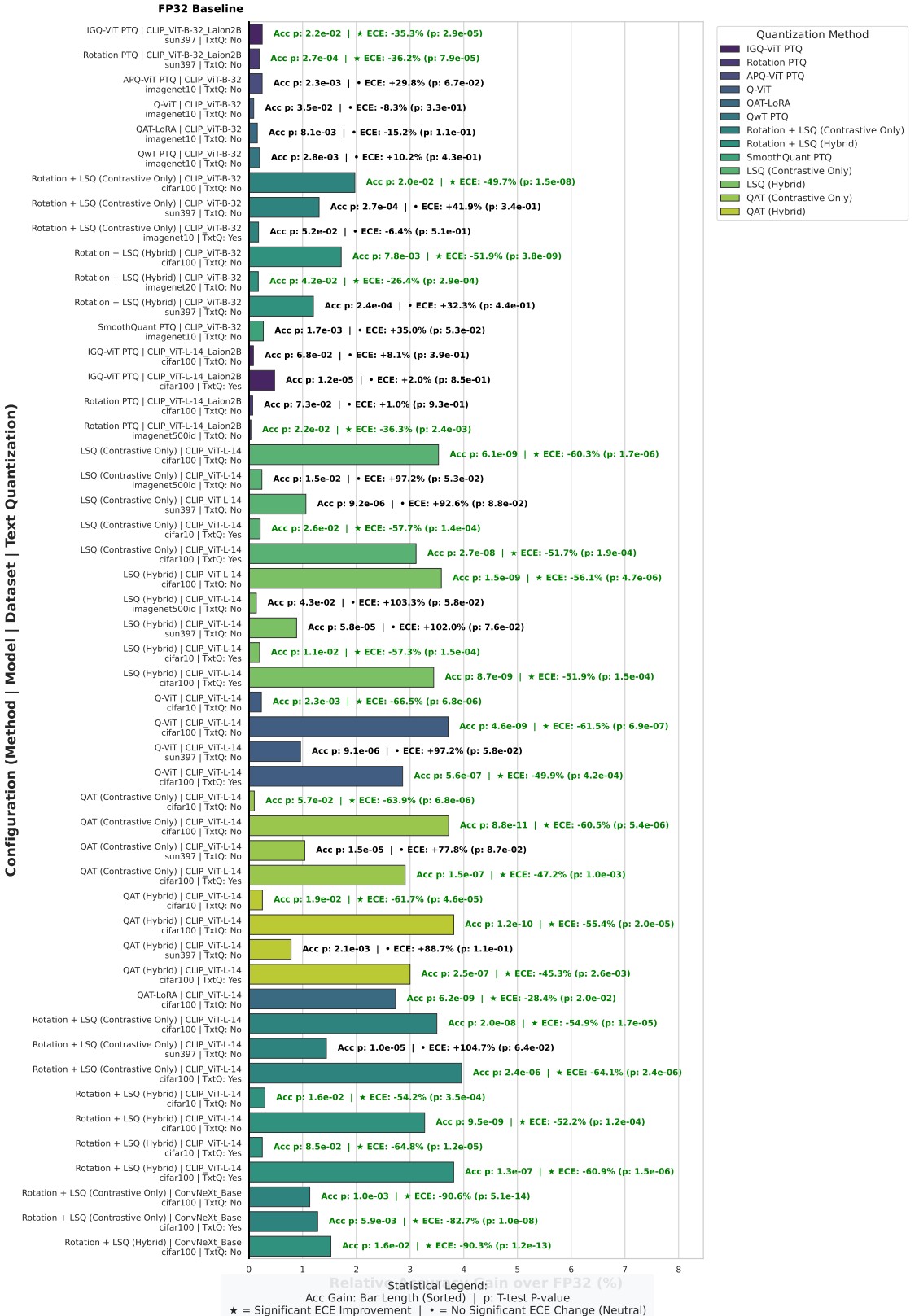

*Figure 26.* Accuracy-Winning configurations (showing only 50). Green text indicates a simultaneous improvement in calibration. While advanced PTQ methods (IGQ, ROT) often show only slight improvement, QAT methods improve accuracy less frequently but more drastically.

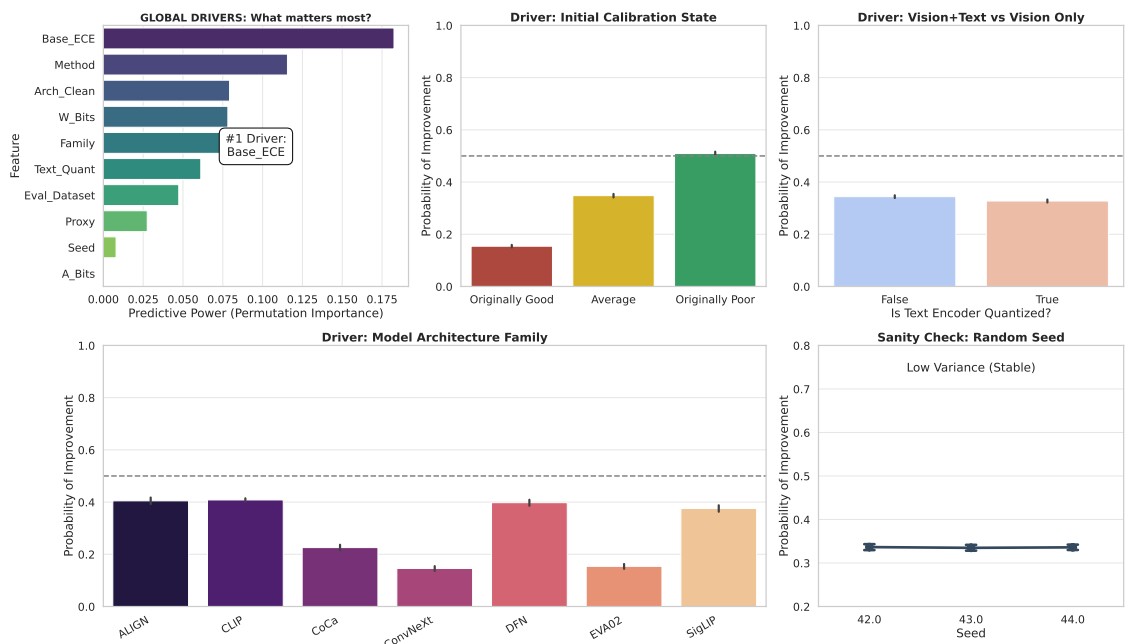

*Figure 27.* What parameters improve calibration? (global view)

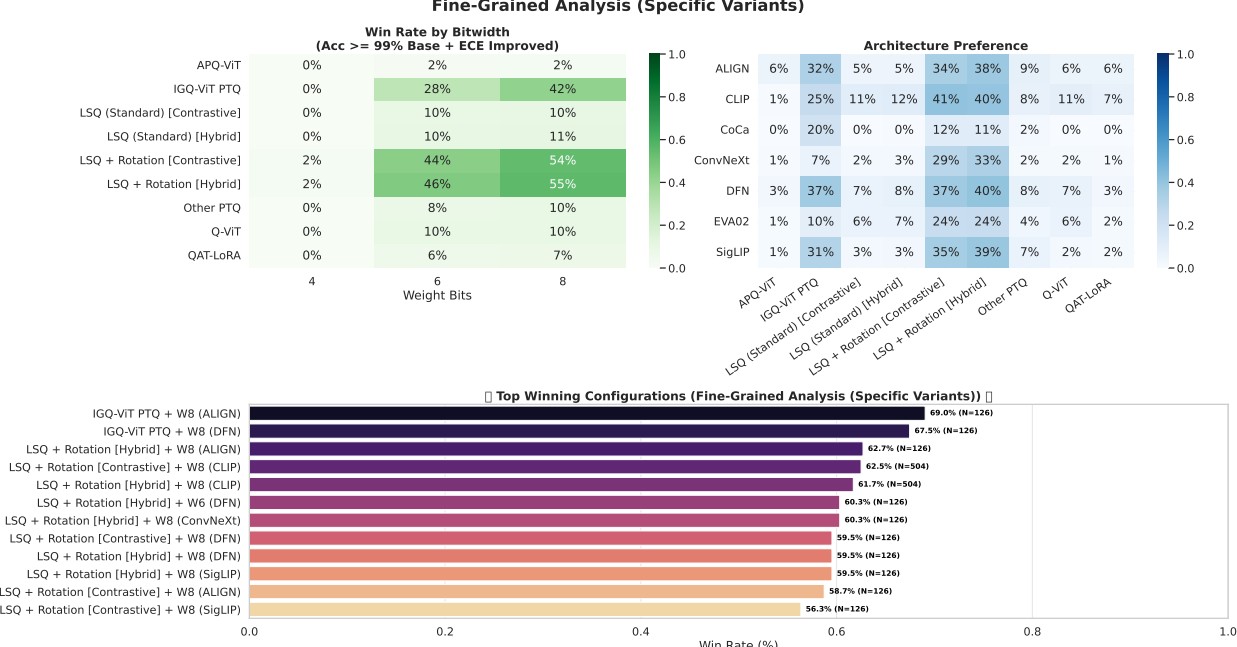

*Figure 28.* What parameters improve calibration? (Closer view)

