# OpenReview forum: "Less Precise Can Be More Reliable: A Systematic Evaluation of Quantization’s Impact on VLMs Beyond Accuracy"
_ICML.cc/2026/Conference — ICML 2026 regular_

### Official Review · Reviewer_U8AB · 2026-03-04

**Soundness:** 3
**Presentation:** 3
**Significance:** 4
**Originality:** 4
**Overall Recommendation:** 5
**Confidence:** 4

**Summary:**

This paper presents a highly detailed and systematic evaluation of VLM quantization. It comprehensively investigates how quantization affects model reliability, including robustness to quantization noise, calibration quality, OOD detection, distribution shift robustness, and susceptibility to spurious correlations. Based on more than 700k+ experiments across over 8k+ quantized models, the paper reports a key finding: quantization is not merely an efficiency-oriented compression technique, but effectively acts as a form of spectral filtering. While this filtering behavior improves calibration and OOD detection performance, it simultaneously makes the model more vulnerable to distribution shifts and spurious correlations.

**Compliance With Llm Reviewing Policy:**

Affirmed.

**Ethical Review Concerns:**

Overall, this work involves a significant amount of work and effort. By conducting in-depth analyses of quantization methods, it offers valuable guidance for future research.

**Final Justification:**

Overall, this work involves a significant amount of work and effort. By conducting in-depth analyses of quantization methods, it offers valuable guidance for future research.

**Key Questions For Authors:**

1. The evaluation in this paper mainly focuses on discriminative VLMs, such as CLIP. However, the multimodal field is rapidly shifting toward generative models based on autoregressive decoding, such as LLaVA and Qwen-VL. Given the fundamental differences between discriminative and generative models in reasoning mechanisms, error accumulation, and hallucination risks, it remains unclear whether the reliability improvements attributed to quantization would generalize to the generative paradigm.

2. While the proposed “spectral filtering” mechanism offers an intuitive explanation for the observed phenomena, it lacks rigorous theoretical formalization. The current argument is largely heuristic and relies on post-hoc SVD analyses. To strengthen the theoretical contribution, it would be important to clarify whether this filtering effect is a distinctive consequence of quantization or a more general form of regularization.

3. Although the paper systematically analyzes the complex impact of quantization on model reliability, it would be valuable to further elaborate on the practical engineering implications of these findings. For example, the authors could discuss how the observed improvement in calibration under quantization might inform model selection or deployment strategies in safety-critical applications.

**Limitations:**

Yes

**Strengths And Weaknesses:**

**Soundness**

This paper presents an in-depth analysis of the impact of quantization on VLMs across multiple reliability dimensions, including accuracy, calibration, OOD detection, distribution shift robustness, and susceptibility to spurious correlations. The study is based on more than 700k+ evaluation runs over 8k+ quantized models, demonstrating an impressive experimental scale. However, the analysis primarily focuses on non-generative (discriminative) VLMs such as CLIP, with limited discussion of mainstream generative vision-language models, such as LLaVA. As a result, the generalizability of the findings to generative VLMs remains unclear.

**Presentation**

The paper is logically structured, progressing coherently from the experimental setup to multi-dimensional empirical analysis and ultimately to a mechanistic interpretation. However, given the large number of experimental dimensions, the appendix contains an extensive amount of data, which may be difficult to navigate. It would be helpful to include a concise summary table in the main text that compares the strengths and weaknesses of different quantization methods across the five reliability dimensions, enabling readers to quickly grasp the key findings.

**Significance**

This paper challenges the conventional belief that quantization inevitably degrades model performance, demonstrating strong academic value and practical significance. Its core contribution lies in revealing that quantization is not merely a compression technique, but can function as a form of spectral filtering with implicit regularization effects. These findings offer a novel perspective on the trade-off between efficiency and reliability, providing meaningful guidance for resource-constrained and edge deployment settings.

**Originality**
Although quantization and model reliability have each been studied independently, systematically integrating the two and conducting a large-scale cross-evaluation specifically for the complex setting of visual language models (VLMs) represents a novel and meaningful contribution. Furthermore, the paper introduces a “spectral filtering” perspective to interpret the regularization effect of quantization, offering a fresh conceptual framework that may inspire the design of more reliable and robust models for edge deployment.

---

> ### Author Rebuttal · Authors · 2026-03-31
>
> We sincerely thank Reviewer U8AB for the detailed and constructive evaluation, and for highlighting the significance and originality of our work.
>
> ---
> ### Q1. Generalizability to Generative VLMs
> We acknowledge this limitation in scope, which is explicitly discussed in **Footnote 1** and **Appendix A**. Because our study was becoming too big, we chose to focus on CLIP-family models because they serve as the vision encoders in larger generative VLMs (e.g., LLaVA uses CLIP ViT-L/14) but also more recent works [1][2][3][4] etc. Our findings about how quantization reshapes the visual encoder's spectral properties and the resulting reliability trade-offs should directly inform decisions about quantizing the vision backbone in these larger systems. We view this as a necessary first step and leave direct generative VLM evaluation as future work. We will explicitly state in our limitations that, while our results should translate to upstream use cases, we did not directly test any.
>
> [1] Driess et al., "Knowledge Insulating Vision-Language-Action Models: Train Fast, Run Fast, Generalize Better," NeurIPS (Spotlight), 2025.
>
> [2] Zeng et al., "FutureSightDrive: Thinking Visually with Spatio-Temporal CoT for Autonomous Driving," NeurIPS (Spotlight), 2025.
>
> [3] Zhang et al., "4D-VLA: Spatiotemporal Vision-Language-Action Pretraining," NeurIPS, 2025.
>
> [4] Zhou et al., "ChatVLA-2: Vision-Language-Action Model with Open-World Reasoning," NeurIPS, 2025.
>
> ---
> ### Q2. Theoretical Formalization of Spectral Filtering
>
> Standard regularizers (Decay, L2, augmentation, Cutout, dropout) operate as continuous, training-time constraints that penalize weight magnitudes or zero out random activations to inject stochastic noise. A more relevant comparison is spectral normalization [1], which explicitly targets the spectrum by dividing W by σ₁(W). However, SN also increases spectral bias but preserves the relative structure of the singular value spectrum, leaving the fine-grained spectral tail intact.
>
> Unlike other regularization techniques, quantization imposes hard representational constraints by restricting values to a finite set of levels with step size Δ, creating an absolute noise floor of Δ²/12 determined solely by the bit-width. As noted in Appendix C.1 (Eqs. 5–7), the per-rank SQNR is: SQNR_k = 10 · log₁₀(σ_k² / (Δ²/12))
>
> Since σ_k decays with rank, components beyond a critical rank k* fall below this floor and are effectively destroyed. Where SN preserves the spectral tail through uniform scaling, quantization obliterates it from the bottom up, thus reducing sensitivity to fine-grained detail and forcing reliance on general features. This discrete, rank-dependent cutoff has no continuous regularization analogue.
>
> For example, we empirically verify by comparing W8A8 quantization with dropout on the same architecture. Dropout reduces spectral bias by distributing energy across ranks [2]; quantization amplifies it by selectively destroying high-rank components (code added in the anonymized repository: https://anonymous.4open.science/r/anno_repo_258741-8F8A/README.md):
>
> Overall, this filtering effect is distinct from quantization but shares similarities with other regularization techniques. A formal theoretical characterization of these connections is left for future work.
>
> [1] Miyato et al., "Spectral Normalization for Generative Adversarial Networks," ICLR, 2018.
>
> [2] Cao et al., "Towards Understanding the Spectral Bias of Deep Learning," IJCAI, 2021.
>
> [**Figure R1: Spectral Energy Shift under W8A8 Quantization vs. Dropout https://imgur.com/a/B1DU7X9 **](https://imgur.com/a/B1DU7X9)
>
> ---
>
> ### Q3. Practical Engineering Implications
> We will add a "Deployment Guidelines" subsection to Section 6. Concretely, practitioners in safety-critical domains (medical imaging, autonomous driving, etc.) where knowing *when the model is uncertain* matters as much as raw accuracy can use Rotation+LSQ to simultaneously improve calibration (lower ECE), accuracy, and noise robustness (**Table 1, Figure 14**), while gaining inference speedup.
>
> We also acknowledge the recommendation in the presentation section of the review and agree that the paper's density hinders practical recommendations per method for practitioners. We will add a concise summary table to the revised main text, mapping each method family to its effect on each of the five reliability dimensions (accuracy, calibration, etc.), along with each quantization method's advantages and complexity. We will also iterate on the Appendix navigation based on reader feedback.
>
> We also validated this with **real TensorRT INT8 deployment**: LSQ INT8 improves over FP16 (+1.86 pp, p < 10⁻⁷), with 1.45× speedup and 34% model size reduction. Blur robustness also improves by 2.2–4.9 pp across severity levels (all p < 10⁻⁸). These results confirm that the reliability benefits we characterize are not simulation artifacts and transfer directly to deployment. More details in our rebuttal to reviewer @5UH2.

---

> > ### Author Rebuttal · Reviewer_U8AB · 2026-04-01
> >
> > The authors have addressed my concern. I will increase my score from 4 to 5.

---

> > > ### Author Response · Authors · 2026-04-02
> > >
> > > We are thankful for the positive feedback and reconsideration of the score. We especially thank the reviewer for highlighting the significance and originality of our work and for acknowledging our rebuttal.

---

### Official Review · Reviewer_Upyg · 2026-03-12

**Soundness:** 3
**Presentation:** 3
**Significance:** 3
**Originality:** 3
**Overall Recommendation:** 5
**Confidence:** 2

**Summary:**

A large-scale empirical study showing that quantization systematically affects multiple reliability properties of VLMs, and proposing a spectral filtering interpretation for these effects.

**Compliance With Llm Reviewing Policy:**

Affirmed.

**Final Justification:**

This paper conducts a large‑scale empirical study on how quantization influences various reliability properties of vision‑language models, and proposes a unified spectral filtering interpretation. The work stands out for its extensive, well‑designed experiments across common quantization methods and models, and it goes beyond simple top‑1 accuracy to analyze calibration, robustness, covariate shift, and spurious correlations. These contributions are original, practically meaningful, and well‑presented.

In my initial review, I raised questions about the conceptual mismatch between dropout and quantization, the lack of controlled comparisons across model architectures and sizes under the same dataset, and the absence of causal experiments to validate the spectral mechanism. I also noted the extremely high correlation between OOD detection and classification accuracy and asked whether they measure the same underlying property.

The authors responded clearly and concretely: they explained that the high correlation reflects shared model redundancy rather than similar mechanisms; they provided valid cross‑architecture and cross‑size comparisons; and most importantly, they added direct spectral intervention experiments using SVD rank truncation to strengthen the causal reasoning. The high correlation between OOD and accuracy is also well explained as coupled degradation, not redundant metrics.

As a result, the soundness of the paper is now noticeably improved, while originality, significance, and presentation remain strong. The rebuttal does not just answer my comments—it meaningfully strengthens the paper.

I therefore upgrade my score. The paper is technically solid, offers valuable insights, and deserves publication.

**Key Questions For Authors:**

1. The correlation coefficient between OOD detection and classification accuracy reaches 0.994. Does this imply that the two metrics essentially reflect different views of the same underlying property?

**Limitations:**

yes

**Strengths And Weaknesses:**

# Strength
1. The paper presents very solid experimental results. The authors support their claims with extensive PTQ and QAT experiments covering mainstream quantization approaches and common models, making this work a valuable summary with thorough theoretical analysis.
2. The work adopts a broad perspective, going beyond the common focus on top-1 classification accuracy in quantization research. It analyzes the practical effects of quantization from a fresh viewpoint. Instead of simply praising quantization, the authors clearly analyze its pros and cons: calibration and noise robustness may improve, while covariate shift and spurious correlation tend to degrade. The depth at this level is sufficient.
3. The work convincingly shows that quantization affects more than just accuracy. The problem formulation is reasonable and the evaluation framework is well-designed. As a paper based on large-scale benchmarking and unified summarization of phenomena, it is largely successful.
4. The paper systematically summarizes several well-known observations under quantization for the first time, such as accuracy improvement caused by quantization, and supports them with sufficient experimental samples.
# Weaknesses
1. In Section 4.1, Fig. 2 shows a strong correlation (r=0.91) between quantization degradation and weight dropout sensitivity. However, dropout corresponds to random parameter perturbation, while the authors describe quantization as systematic feature filtering. Is there a potential mismatch between these two mechanisms?
2. Section 4.2 claims that models with higher redundancy and better training quality are easier to quantize. The authors only compare the same model trained on datasets of different quality, but lack comparisons across different model architectures/sizes on the same dataset. Although this may seem intuitive, a theoretically oriented summary paper requires further justification and verification.
3. The theoretical proposal in Section 5 interprets quantization as a filter, but lacks interventional experiments to validate the causal chain. For instance, the authors do not verify by manipulating spectral components directl. e.g., removing only high-rank components to check whether reliability changes match quantization effects, or retaining only high-rank components to see if covariate shift degradation is alleviated. As a result, the proposed mechanism remains a plausible explanation rather than rigorous causal evidence.

---

> ### Author Rebuttal · Authors · 2026-03-31
>
> We sincerely thank Reviewer Upyg for recognizing the depth and breadth of our experimental analysis and for the constructive feedback.
>
> ---
>
> ### W1. Dropout vs. Quantization Mismatch (Section 4.1)
>
> The correlation exists because both operations probe the same underlying property: **parameter redundancy**. A model whose accuracy collapses under random weight dropout has low structural margin and the same lack of margin makes it brittle to quantization's discrete truncation. The r=0.91 correlation therefore reflects a shared *diagnostic* (redundancy), not a shared *mechanism*.
>
> While we agree that both are regarded as regularization techniques that inject noise, they remain distinct, which we confirm empirically with a **spectral comparison**:
>
> [**Link to Figure R1: Spectral Energy Shift under W8A8 Quantization vs. Dropout https://imgur.com/a/B1DU7X9 **](https://imgur.com/a/B1DU7X9)
>
> We will add a clarifying sentence in Section 4.1 distinguishing the shared diagnostic from the distinct mechanisms.
>
> ---
>
> ### W2. Model Capacity vs. Quantization Difficulty (Section 4.2)
>
> We generally compare pre-training data quality, but we also provide controlled comparisons across model sizes on the same dataset:
>
> - **Controlling for LAION-2B:** `ViT-B-32` vs. the larger `ViT-L-14`, both trained on `LAION-2B`. The larger model shows significantly better quantization resilience (**Section 4.1, Lines 199–203; Figure 12**).
> - **Controlling for WIT-400M:** `ViT-B-32` vs. `ViT-L-14`, both on `WIT-400M`. The same conclusion holds (when including QAT methods, this is overall the best "quantizable" model due to increased model size and data curation).
> - **Cross-architecture at Base scale:** `ViT-B/32`, `ConvNeXt-Base`, `EVA02-B-16`, and `SigLIP-B-16`, `ALIGN-B-32`, all on LAION-scale data.
>
> We acknowledge that a fully factorial design (multiple architectures × multiple sizes × same dataset) would improve our results, but it would also require retraining foundation models from scratch, which is beyond our evaluation-focused scope. We will add this to our limitations section.
>
> ---
>
> ### W3. Causal Evidence for Spectral Filtering (Section 5)
>
> We performed high-rank truncation experiments on energy, noise, and accuracy, and we agree with the reviewer that demonstrating the causal chain (linking spectral bias exacerbation caused by quantization to downstream reliability effects) is necessary to qualify our work as rigorous.
>
> We extended this intervention on CIFAR-100 and ImageNet-1k, from accuracy to calibration, noise robustness, and spurious correlations (code added in the anonymized repository: https://anonymous.4open.science/r/anno_repo_258741-8F8A/README.md):
>
> [**Figure R2: SVD Rank Truncation on CIFAR-100 and ImageNet-1k https://imgur.com/a/ibXIAeF **](https://imgur.com/a/ibXIAeF)
>
> *Note the increased effect at lower ranks.*
>
> [**Figure R3: Spurious Degradation vs. SVD Truncation Rank https://imgur.com/a/x2BCyVt **](https://imgur.com/a/x2BCyVt)
>
> This confirms that spurious amplification originates specifically from the loss of high-rank discriminative features. We will add these results to the revised Section 5 or the Appendix.
>
> We also conducted the complementary experiment, inferring only from high-rank components while truncating out the dominant low-rank subspace. However, since the lower ranks concentrate the vast majority of the representational energy, removing them leaves low-energy features and noise, making meaningful downstream inference challenging. e.g., using the top 100-768 ranks yields less than 10% accuracy on ImageNet, making any downstream comparison unreliable.
>
> ---
>
> ### Q1. OOD Detection and Accuracy Correlation (r = 0.994)
>
> While prior work [1] has indeed established a strong global correlation (r = 0.994) between OOD detection and classification accuracy, this relationship was historically observed across different models as their baseline accuracy naturally improved (i.e., different models with stronger zero-shot accuracy also exhibited better OOD AUROC and FPR@95). Our objective was to investigate whether this tight coupling persists in the reverse scenario: within the same model when its accuracy is actively degraded by quantization. As shown in Appendix F (Figure 16A), this correlation is largely maintained across most architectures under compression.
>
> We agree the correlation is striking, but it reflects a shared sensitivity to representation degradation under quantization rather than metric redundancy. For example, miscalibrating a model would severely degrade its OOD detection performance without reducing its accuracy under most OOD detection methods. It is also noteworthy that some of the OOD detection methods we used rely on the top-1 zero-shot accuracy of VLMs (such as NegLabel).
>
> [1] Wang, Q., Lin, Y., Chen, Y., Schmidt, L., Han, B., & Zhang, T. (2024). A sober look at the robustness of CLIPs to spurious features. In *Advances in Neural Information Processing Systems (NeurIPS)*.

---

> > ### Author Rebuttal · Reviewer_Upyg · 2026-04-02
> >
> > Thank you for your detailed and responsive rebuttal. Your clarifications and additional experiments have adequately addressed my concerns. I will raise my score accordingly.

---

> > > ### Author Response · Authors · 2026-04-02
> > >
> > > We thank the reviewer for their feedback and for raising their score. We are encouraged that the additional experiments were helpful and that they addressed their concerns.
> > >
> > > We would also like to use this response opportunity to issue a brief erratum regarding our rebuttal to Q1: We accidentally cited Wang et al. ("A sober look at the robustness of CLIPs to spurious features", NeurIPS 2024) instead of Tu et al. ("A Closer Look at the Robustness of Contrastive Language-Image Pre-Training", NeurIPS 2023) due to their highly similar titles and venues. We also want to clarify that while Tu et al. established this strong correlation, they did not provide a global correlation score; the specific $r = 0.994$ figure in the first sentence is the accuracy/OOD AUROC degradation due to quantization correlation score was the score that we found, and was leftover from a previous rebuttal draft.

---

### Official Review · Reviewer_5UH2 · 2026-03-13

**Soundness:** 3
**Presentation:** 3
**Significance:** 4
**Originality:** 3
**Overall Recommendation:** 5
**Confidence:** 4

**Summary:**

This paper presents a large-scale empirical study of the impact of quantization on the reliability of vision–language models (VLMs). The authors evaluate 16 quantization techniques across 10 pretrained VLMs and measure their effects on five types of reliability and robustness metrics, including calibration, out-of-distribution detection, and robustness to distribution shifts.
A key observation that emerges from these experiments is that quantization often behaves like a low-pass spectral filter. Under this interpretation, quantization suppresses high-frequency components of the learned representation and effectively regularizes the model. As a result, the model tends to rely more heavily on higher-level features, which can improve certain robustness properties.
Beyond this central hypothesis, the paper provides systematic empirical analyses of several phenomena related to quantized VLMs. These include comparisons between post-training quantization (PTQ) and quantization-aware training (QAT), the importance of cleaner pretraining data for quantizable models, differences between synthetic and semantic robustness, and evidence that quantization can increase reliance on spurious correlations.
Although the paper does not introduce a new quantization method, it provides a comprehensive and careful empirical study that offers new insights into how quantization affects reliability in multimodal models.

**Compliance With Llm Reviewing Policy:**

Affirmed.

**Final Justification:**

This is a strong and interesting paper. The additional results shared during the rebuttal nicely round out the analysis. I will maintain my score.

**Key Questions For Authors:**

I think this paper is solid and an important style of work. My only questions would be to address the points listed in the weaknesses section.

**Limitations:**

yes

**Strengths And Weaknesses:**

# Strengths
## Soundness
The paper is technically sound and provides a large-scale empirical study of quantization effects on VLM reliability. The evaluation spans 16 quantization methods and 10 VLM models, covering several reliability metrics including calibration, out-of-distribution detection, robustness to synthetic corruptions, semantic robustness, and spurious correlations. The experimental setup is comprehensive and the analyses are systematic. The paper also proposes a spectral filtering interpretation of quantization, supported by empirical analysis of representation structure, which helps explain several observed behaviors such as improved robustness to synthetic corruptions and changes in calibration.

## Presentation
The paper is generally well written and clearly structured. The authors organize the results around several clear empirical observations, including comparisons between PTQ and QAT, the role of training data quality in quantization robustness, and differences between synthetic and semantic robustness. The spectral filtering hypothesis provides a useful conceptual framework that helps unify the empirical findings.

## Significance
Understanding how quantization affects reliability properties of VLMs is an important and timely problem. Quantization is widely used for efficient model deployment, yet most prior work focuses primarily on accuracy preservation rather than broader reliability considerations. This paper expands the evaluation of quantized models to include calibration, robustness, and spurious correlations, which are critical properties in real-world deployment settings. The breadth of experiments and systematic analysis make this work a useful reference for future research on model compression and robustness.

## Originality
Although the paper does not introduce a new quantization method, it provides originality through large-scale empirical analysis and new insights into how quantization affects representation structure and reliability. The interpretation of quantization as a form of spectral filtering that acts as a regularizer is a particularly interesting perspective. The work also highlights several non-obvious phenomena, such as the differing effects of quantization on synthetic versus semantic robustness and the increased reliance on spurious correlations.

# Weaknesses
## Soundness
N/A

## Presentation
1. Figure 1 is slightly confusing, there may be a better way to present results that is immediately obvious to the reader.
2. It could be useful to add a comparison of efficiency metrics between different quantization methods.

## Significance
N/A

## Originality
N/A

---

> ### Author Rebuttal · Authors · 2026-03-31
>
> We thank Reviewer 5UH2 for the positive assessment and for recognizing the technical soundness, significance, and originality of our work.
>
> ---
>
> ### W1. Presentation of Figure 1
>
> > *"Figure 1 is slightly confusing; there may be a better way to present results that is immediately obvious to the reader."*
>
> We agree. In the revision, we will: (i) add explicit quadrant labels (*Win–Win*, *Compromise*, *Degradation*) directly on the figure, (ii) increase font sizes for axis labels and annotations, and (iii) improve interpretability with more illustrative arrows. We will iterate with first-time readers to ensure the teaser figure is immediately easy to digest and effectively introduces the paper.
>
> ---
>
> ### W2. Efficiency Metrics Comparison for Enhancing the Presentation
>
> > *"It could be useful to add a comparison of efficiency metrics between different quantization methods."*
>
> A quantization method's efficiency largely depends on framework-level support. Layers that lack available conversion strategies or kernel support silently fall back to FP16/FP32, drastically reducing the effective speedup from the theoretical maximum; for instance, using Rotation techniques will make the engine silently fall back to FP32 for most layers. The gains are also model-dependent, as ViT-B/16 achieves a 1.9× speedup, while ViT-B/32 achieves only a 1.45× speedup. We will include in the appendix a general comparison table between quantization methods using several criteria.
>
> To validate that our simulated findings hold in practice, we deploy CLIP ViT-B/32 (WIT) as **real TensorRT INT8 engines** with QDQ node placement on an NVIDIA RTX 4070. Results are averaged over **10 random seeds** with standard deviations reported:
>
> Note: FP16 inference always preserves FP32 performance.
>
>
>
> **Table R1.** Clean accuracy and inference efficiency on CIFAR-100 (n=10 seeds, BatchSize=64).
>
> | Precision | Clean Acc (%) | Δ vs FP32 (pp) | Latency (ms/batch) | Speedup vs FP32 | Speedup vs TRT FP16 | Size |
> |---|---|---|---|---|---|---|
> | FP32 PyTorch | 64.47 | — | 120.89 | 1.00× | — | 302.1 MB²  |
> | TRT FP16 | 64.43 ± 0.05 | −0.04 | 22.70 | 5.33× | 1.00× | 178.7 MB |
> | TRT INT8 (LSQ) | **66.30 ± 0.39** | **+1.83** | **15.66** | **7.72×** | **1.45×** | **118.4 MB** |
>
> ²Assuming text embeddings are pre-computed and saved
>
> The LSQ INT8 engine exceeds FP32 accuracy by **+1.83 pp** (t=14.98, p < 10⁻⁷) and FP16 by **+1.86 pp** (t=15.23, p < 10⁻⁷). vs. PyTorch FP32: **7.72× faster, 5.1× smaller**. vs. TRT FP16: **1.45× faster, 34% smaller** (118.4 vs 178.7 MB).
>
> **Table R2.** Blur robustness on real TensorRT engines (n=10 seeds).
>
> | Blur Condition | FP16 Acc (%) | INT8 Acc (%) | FP16 Drop (pp) | INT8 Drop (pp) | TRT INT8 Added Rob. (pp) | p-value |
> |---|---|---|---|---|---|---|
> | Mild (k=11, σ=2) | 60.51 ± 0.07 | 64.59 ± 0.37 | 3.92 ± 0.11 | 1.71 ± 0.36 | **2.21 ± 0.33** | < 10⁻⁸ |
> | Moderate (k=23, σ=3.8) | 52.15 ± 0.02 | 58.31 ± 0.45 | 12.28 ± 0.08 | 7.99 ± 0.41 | **4.29 ± 0.36** | < 10⁻¹⁰ |
> | Severe (k=51, σ=8) | 28.79 ± 0.04 | 35.58 ± 0.79 | 35.64 ± 0.07 | 30.72 ± 0.64 | **4.92 ± 0.60** | < 10⁻⁸ |
>
> All differences are statistically significant (paired t-test, p < 10⁻⁸). The TRT INT8 model is not only more accurate but also **consistently more robust** to blur corruption and the advantage grows with severity (+2.21 pp at mild → +4.92 pp at severe). This confirms that the reliability improvements reported with simulated quantization **transfer directly to real deployed INT8 engines**.
>
> We added the code for the end-to-end zero-shot OpenCLIP ViT-B/32 (WIT) TensortRT conversion and benchmark on CIFAR-100 in the fully anonymized repository: https://anonymous.4open.science/r/anno_repo_258741-8F8A/README.md (`lsq_cifar100_trt.py`)
>
> Note: Converting a fake-quantized model to a TensorRT engine while preserving performance and adhering to its Q/DQ scheme requires substantial architecture-specific effort, as the process is model-locked and may not support other models. Our study uses simulated quantization, which faithfully models real precision loss; we validated this by successfully deploying select configurations as TensorRT engines, though framework-level optimizations may introduce minor numerical deviations. To our knowledge, this is the first work to quantize CLIP-family models to W8A8 precision using QAT methods and that achieves a significant speedup. This functionality was recently added, and CLIP W8A8 was not supported by TensorRT until very recently, and was largely silently falling back to FP16/FP36.

---

> > ### Author Rebuttal · Reviewer_5UH2 · 2026-04-03
> >
> > Thanks to the authors for addressing my questions. I'm happy to maintain my score.

---

> > > ### Author Response · Authors · 2026-04-07
> > >
> > > We are very grateful for the reviewer's positive feedback and for their review of our work. We are particularly encouraged by the reviewer's recognition of the significance of our contributions.

---

### Decision · Program_Chairs · 2026-04-30

**Decision:**

Accept (regular)

**Comment:**

This paper presents a systematic study of the impact of quantization on the reliability of vision-language models (VLMs). Going beyond conventional evaluations that primarily focus on top-1 accuracy, the authors conduct a large-scale study with over 700k evaluation runs under diverse configurations, examining the effects of quantization across a wide range of reliability-related metrics. A key finding is that, contrary to the common intuition that quantization degrades performance, it can in some cases improve accuracy, OOD detection, and robustness to noise. At the same time, it does not mitigate issues such as covariate shift or spurious correlations. To explain these observations, the authors propose a spectral filtering perspective, suggesting that quantization attenuates high-frequency components and promotes reliance on more robust low-rank features. This interpretation is further supported by experiments showing that explicit spectral filtering induces effects consistent with those observed under quantization.

All three reviewers recommend acceptance, with largely consistent evaluations. In particular, they highlight the primary contributions of this paper: a comprehensive empirical study across diverse models, configurations, and reliability metrics, which reveals previously underexplored effects of quantization and offers a unifying interpretation through the spectral filtering perspective. While initial reviews pointed out issues related to clarity and the depth of analysis, the authors addressed them in the author response with additional explanations and experimental evidence, which the reviewers found satisfactory.

Overall, the paper presents a solid and extensive empirical study that provides novel and well-supported insights, making it a valuable contribution to the community. One remaining concern is its practical impact. As noted by Reviewer U8AB, the experiments are limited to classification-style VLMs (e.g., CLIP and its variants). Compared to multimodal LLM-based VLMs (e.g., LLaVA), these models are relatively lightweight, and quantization is typically less critical. As a result, the broader applicability and practical impact of the findings may be somewhat limited.

Last but not least, the following reference in the paper may be fictitious and needs to be corrected:
Tallec, C., Blier, L., and Ollivier, Y. Revisiting the regularization effect of quantization. arXiv preprint arXiv:2310.03113, 2023.